# Revisiting species and areas of interest for conserving global mammalian phylogenetic diversity

Marine Robuchon [1,2,3 ✉], Sandrine Pavoine [1], Simon Véron[4], Giacomo Delli[3], Daniel P. Faith[5], Andrea Mandrici[3], Roseli Pellens [4], Grégoire Dubois [3] & Boris Leroy [2]

Various prioritisation strategies have been developed to cope with accelerating biodiversity loss and limited conservation resources. These strategies could become more engaging for decision-makers if they reflected the positive effects conservation can have on future projected biodiversity, by targeting net positive outcomes in future projected biodiversity, rather than reflecting the negative consequences of further biodiversity losses only. Hoping to inform the post-2020 biodiversity framework, we here apply this approach of targeting net positive outcomes in future projected biodiversity to phylogenetic diversity (PD) to re-identify species and areas of interest for conserving global mammalian PD. We identify priority species/areas as those whose protection would maximise gains in future projected PD. We also identify loss-significant species/areas as those whose/where extinction(s) would maximise losses in future projected PD. We show that our priority species/areas differ from loss-significant species/areas. While our priority species are mostly similar to those identified by the EDGE of Existence Programme, our priority areas generally differ from previously-identified ones for global mammal conservation. We further highlight that these newly-identified species/areas of interest currently lack protection and offer some guidance for their future management.

[1] Centre d'Ecologie et des Sciences de la Conservation (CESCO), Muséumnational d'Histoire naturelle, Centre National de la Recherche Scientifique, Sorbonne Université, Paris, France. [2] Biologie des Organismes et Ecosystèmes Aquatiques (BOREA), Muséum national d'Histoire naturelle, Centre National de la Recherche Scientifique, Institut de Recherche pour le Développement, Sorbonne Université, Université Caen-Normandie, Université des Antilles, Paris, France. [3] Joint Research Centre (JRC) of the European Commission, Directorate for Sustainable Resources, Ispra (VA), Italy. [4] Institut de Systématique, Evolution, Biodiversité (ISYEB), Muséum national d'Histoire naturelle, Centre National de la Recherche Scientifique, Sorbonne Université, Ecole Pratique des Hautes Etudes, Université des Antilles, Paris, France. [5] Australian Museum, Sydney, Australia. ✉email: marine.robuchon@ec.europa.eu

In an era of biodiversity crisis and limited resources to protect biodiversity, prioritising species and areas for conservation is essential. Traditionally, prioritisation frameworks seek to protect the species that are the most threatened or spatially restricted, and the areas that exhibit the highest species richness[1–4]. Such frameworks may however overlook the evolutionary history that different species embody. Phylogenetic diversity (PD) is commonly used to quantify the evolutionary history represented by a species or a set of species[5] and represents the feature diversity of such a set (a "feature" is any particular trait evolutionary-inherited). Maximising PD is a sound conservation strategy as it maximises biodiversity at the level of features. This feature diversity represents a reservoir of unanticipated future benefits for humanity ("biodiversity option values")[5]. In accord with this, the Intergovernmental Platform on Biodiversity and Ecosystem Services (IPBES) in its global assessment report on biodiversity and ecosystem services[6] adopted PD as an indicator of maintenance of options[7]. Maximising PD may also preserve greater ecosystem functioning[8] and may increase the evolutionary potential of a species set so that one or more members of the set can adapt to changing conditions[5,9], although these statements have been seldom tested and therefore require more empirical evaluations[10–12]. Beyond these utilitarian arguments, maximising PD is also a way to preserve the evolutionary heritage of our planet.

Global PD-based priority species have been identified for mammals. Isaac et al.[13], followed by Collen et al.[14] used the 'evolutionarily distinct and globally endangered' (EDGE) approach to attribute a priority score to a species by combining its score of evolutionary distinctiveness (ED) with its status on the IUCN Red List of Threatened Species™ as an estimate of its extinction risk. While EDGE used a simple partitioning of the total PD of a clade among its member species, species of interest for conserving PD may also be identified based on how our actions on such species – assuming they produce some changes in their probabilities of extinction – influence future projected PD. This framework, which uses probabilities of extinction to calculate future projected PD (hereafter expected PD), can cover any scenario of changes in probabilities of extinction[15]. The 'heightened evolutionarily distinct and globally endangered' (HEDGE) approach[16] is a special case of the expected PD framework. It is particularly relevant to identify PD-based priority species, as the HEDGE score of a species corresponds to the gain in expected PD if the species was secured (probability of extinction goes to 0). Species with the highest HEDGE scores are typically threatened, evolutionarily distinct species. One advantage of priorities based on expected PD such as HEDGE over EDGE priorities is that they reflect the opportunity for averted loss of PD (i.e. safeguard more PD) in a future time horizon, assuming that conservation action on a species produces some nominated reduction in its probability of extinction[17]. The globally important 'EDGE of Existence Programme'[13,18] is endorsing expected PD calculations[17,19] as an alternative to its conventional EDGE scores (https://www.edgeofexistence.org/blog/cutting-edge-updating-science-behind-species/). As another special case of the expected PD framework, Steel et al.[16] introduced a score corresponding to the loss in expected PD if the species were to go extinct (probability of extinction goes to 1). The species exhibiting the highest values of such score are typically secure, evolutionarily distinct and Faith[20] defined 'loss-significant, evolutionarily distinctive, globally enduring' species (LEDGE) to refer to the particular case where extinction of a secure species means a large loss in expected PD. While May-Collado & Agnarsson[21] identified PD-based priority species for aquatic mammals using both EDGE and HEDGE, neither HEDGE nor LEDGE scores have yet been calculated globally for all mammals.

Several approaches have been proposed to identify PD-based priority areas for terrestrial[22–26] or aquatic mammals[27,28]. However, none of these considered all biomes (i.e. terrestrial, freshwater and marine) in the same analysis. Furthermore, among the studies on PD-based priority areas, only one regional study[29] has applied the expected PD framework, using HEDGE and LEDGE scores[17] to identify priority areas.

Despite the increasing awareness that PD conservation would benefit society by maintaining its options and the recognition that it should be explicitly included into global conservation goals[18,30], the preservation of PD is not yet embedded in international biodiversity policies – such as the strategic plan 2011–2020 of the UN Convention on Biological Diversity (CBD)[31] and the new EU biodiversity strategy for 2030[32]. These strategies include evolution explicitly only via the preservation of genetic diversity for cultivated plants, and farmed and domesticated animals. A few months away from the 15th Conference Of the Parties of the CBD which will set the new conservation targets to 2030, the first drafts of the post-2020 biodiversity strategy[33] overlook genetic diversity[34] and do not yet incorporate PD. Recent CBD post-2020 working documents[35] do note that efforts to reduce species extinctions should consider priorities for evolutionarily distinct species across the entire tree of life. Further, they follow a recent proposal[18] that the CBD post-2020 framework adopt the IPBES existing indicator for tracking expected PD loss[7]. However, these proposals do not suggest any indicator that would highlight expected PD gains. To further contribute to promoting PD conservation within the post-2020 framework, we propose to discuss, for the particular case of PD, Bull et al.[36]'s suggestion to move beyond strategies seeking to avoid further biodiversity losses and develop strategies resulting in net positive outcomes for biodiversity. Indeed, strategies targeting net positive outcomes for biodiversity – i.e. biodiversity gains superior to biodiversity losses – would encourage wider engagement in biodiversity conservation[36]. Although the novelty is mainly semantic, using a net positive outcome approach can have major implications for the way in which conservation actions are delivered because it highlights the positive effects conservation actions can have on future projected biodiversity. It offers a more positive vision for the future than imagining the consequences of further biodiversity losses only, and can be applied to PD. We thus expect that targeting net positive outcomes for PD would encourage wider engagement in PD conservation. For example, a prioritisation strategy can be designed to increase gains in expected PD and complemented by preventive conservation actions that would limit losses in expected PD. Here, we use this approach to re-identify species and areas of interest for conserving global mammalian PD.

Specifically, our aim is to re-assess and compare species and areas of interest for conserving global mammalian PD based on the gain in expected PD they can bring if they are protected (priority species/areas) and the loss in expected PD they can trigger if they go extinct (loss-significant species/areas). In contrast with existing PD conservation approaches that focus on maximising current extant PD, the species and areas of interest that we define here focus on maximising gains (priority species/areas) and minimising losses (loss-significant species/areas) of expected PD in a 50-year time horizon (Fig. 1). We then compare the newly identified priorities to previously identified priorities for global mammal conservation. Finally, we examine how the species and areas of interest identified here are currently protected.

## Results and discussion
**Re-identification of species and areas of interest.** In contrast to previous works that identified phylogenetically informed priority species[13,14,17] or areas[15–18] to maximally preserve current

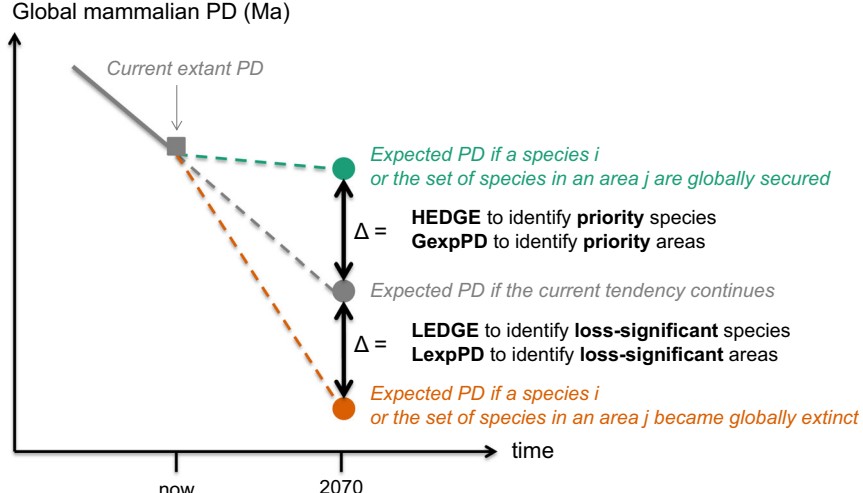

**Fig. 1 Conceptual framework to identify species and areas of interest for conserving global mammalian PD based on the effects our actions can have on expected PD.** (conservation actions leading to species securing = green circle, no change in species trajectories to extinction = grey circle and threatening actions leading to species extinctions = orange circle). Typically, phylogenetically informed conservation priorities have been identified based on the amount of current extant (threatened) PD they represent (grey square), so focussing on averting losses. Here we identified priority species based on the gain in expected PD they can bring if globally secured (HEDGE score) and priority areas based on the gain in expected PD they can bring if all species of the area are globally secured (GexpPD score), in a 50 year time horizon. We also identified loss-significant species based on the loss in expected PD they can trigger if they became globally extinct (LEDGE score) and loss-significant areas based on the loss in expected PD they can trigger if all species of the area became globally extinct (LexpPD score) in a 50-year time horizon.

(threatened) PD, the conservation priorities we identified here correspond to those whose protection would maximise gains in expected PD in the next 50 years (Fig. 1).

The HEDGE scores of the 1369 priority species, i.e. the TOP 25% HEDGE species, range from 0.0535 Ma (for the European rabbit *Oryctolagus cuniculus*) to 33.8 Ma (for the Mountain Pygmy possum *Burramys parvus*) (Supplementary Data 1). These phylogenetically informed priority species belong to 23 out of the 27 mammalian orders and are the most numerous in Rodentia, where they represent 20% of the species. All species of Microbiotheria ($n = 1$), Pholidota ($n = 8$), Proboscidea ($n = 2$) and Sirenia ($n = 4$) are priority species. Figure 2a highlights the 10 species, all threatened, whose protection would bring the most important gains in expected PD.

These HEDGE scores calculated with the phylogeny from PHYLACINE[37,38] are highly correlated to those calculated with the phylogeny of Upham et al.[39] ($\rho = 0.86$, Supplementary Fig. 1). Moreover, 93% of the priority species identified with the phylogeny from Upham et al.[39] are also identified as priority species using the phylogeny from PHYLACINE[37,38]. This suggests that HEDGE scores are robust to recent updates of the mammalian phylogeny.

These 1369 priority species are concentrated in Amazonia, West, Central, East and South Africa, Madagascar and South-East Asia (Fig. 3a) and represent a high proportion of total species richness in the northern parts of Atlantic and Pacific Oceans, and in the Caspian Sea (Supplementary Fig. 2a). This is because a lot of mammal species in the northern parts of Atlantic and Pacific Oceans are priority species, such as the sea otter *Enhydra lutris* (TOP 152 HEDGE), and a lot of mammal species of the Caspian Sea are priority species, such as the Caspian seal *Pusa caspica* (TOP 638 HEDGE).

The priority areas are mainly located in South-East Africa, Madagascar, and South-East and Central Asia (Fig. 4a). They correspond quite well to the hotspots of richness in TOP HEDGE species except for Amazonia (Fig. 3a and Supplementary Table 1), and the gain in expected PD if all species of the cell were secured

is strongly correlated to the richness in TOP HEDGE species ($\rho = 0.89$, Supplementary Fig. 4).

We also identified loss-significant species based on the loss in expected PD they can trigger if they became globally extinct and loss-significant areas based on the loss in expected PD they can trigger if all species of the area became globally extinct in a 50-year time horizon (Fig. 1). The LEDGE scores of the 1369 loss-significant species, i.e. the TOP 25% LEDGE species, range from 3.96 Ma (for the buffy-tufted marmoset *Callithrix aurita*) to 78.6 Ma (for the aardvark *Orycteropus afer*) (Supplementary Data 1). These loss-significant species belong to 26 out of the 27 mammalian orders and are the most numerous in Rodentia where they represent 23% of the species. All species of Dermoptera ($n = 2$), Microbiotheria ($n = 1$), Notoryctemorphia ($n = 2$), Proboscidea ($n = 2$) Sirenia ($n = 4$) and Tubulidentata ($n = 1$) belong to the TOP 25% LEDGE. Figure 2b highlights the 10 species, mostly secured, whose loss would trigger the most important losses in expected PD.

These LEDGE scores calculated with the phylogeny from PHYLACINE[37,38] are only moderately correlated to those calculated with the phylogeny of Upham et al.[39]. ($\rho = 0.54$, Supplementary Fig. 1). Fifty five percent of the loss-significant species identified with the phylogeny from Upham et al.[39] are also identified as loss-significant species using the phylogeny from PHYLACINE[37,38]. This suggests that, contrary to HEDGE scores, LEDGE scores are sensitive to recent updates of the mammalian phylogeny. A possible explanation is that HEDGE scores seem to be more driven by extinction risk than LEDGE scores (Supplementary Fig. 3), and therefore less impacted by phylogenetic changes. With the rapidly evolving phylogenetic knowledge, results will have to be regularly updated and recommendations may evolve accordingly – as this has been suggested and/or done in other studies dedicated to PD conservation[40–42]. In this regard the robustness of HEDGE to phylogeny updates is encouraging because it suggests that priority species will mainly remain the same, even with new phylogenetic knowledge. The sensitivity of LEDGE scores may be less of an issue because these are

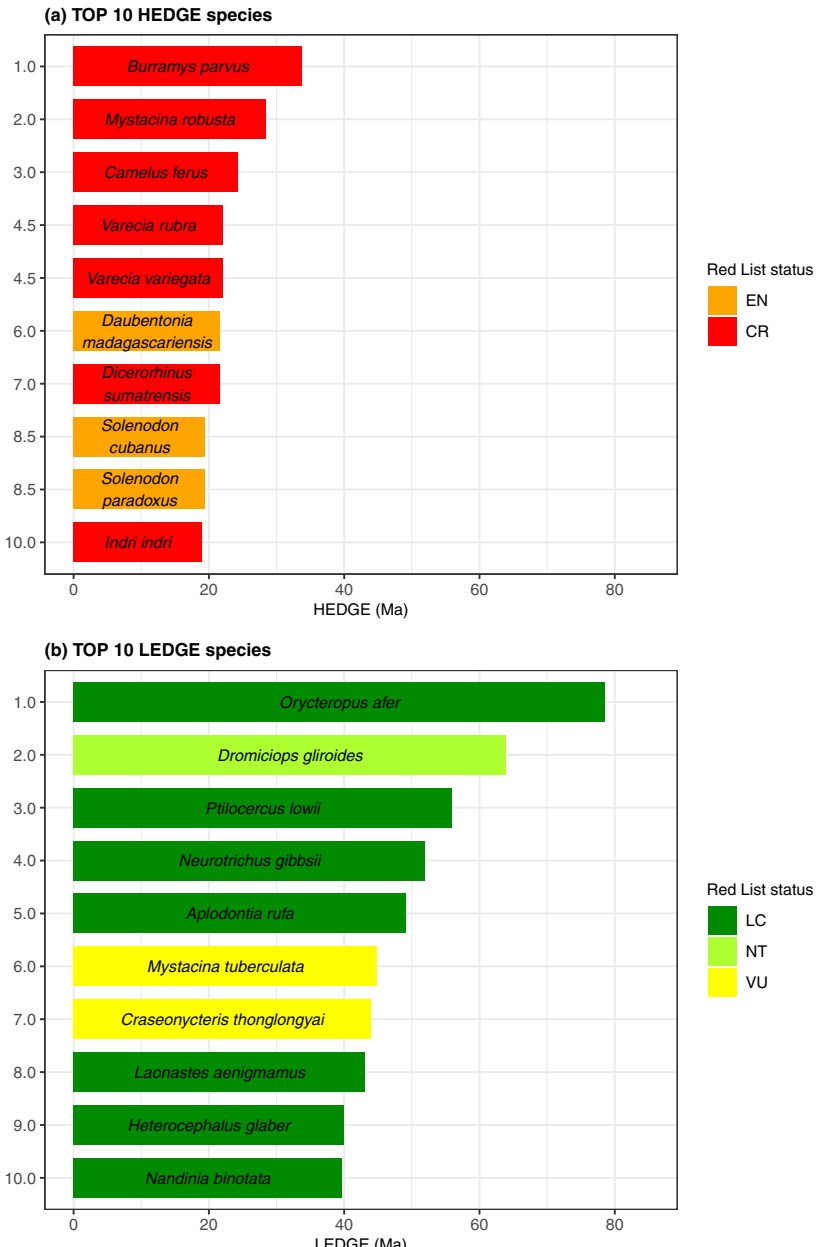

**Fig. 2 Scores and Red List status [LC least concern, NT near threatened, VU vulnerable, EN endangered, CR critically endangered] of some of the species of interest for conserving global mammalian PD.** Scores and Red List status [LC least concern, NT near threatened, VU vulnerable, EN endangered, CR critically endangered] of some of the species of interest for conserving global mammalian PD. Scores and Red List status of the TOP 10 HEDGE species (**a**) and of the TOP 10 LEDGE species (**b**). The species are sorted by their rank (*y*-axis) with top-ranked species at the top. Source data are provided as a Source Data file.

typically secure species, giving us more time to get the phylogeny right.

If we take the PHYLACINE tree as our guide, then the 1369 loss-significant species identified here are concentrated in Central America, Amazonia, Central Africa and South-East Asia (Fig. 3b). They represent a high proportion of total species richness in the northern parts of Atlantic and Pacific Oceans, in the Black Sea, in South America, North Africa, Madagascar and Australia (Supplementary Fig. 2b). This is because a lot of mammal species in these areas are loss-significant species, such as the humpback whale *Megaptera novaeangliae* in the Black Sea (TOP 1119 LEDGE) and the platypus *Ornithorhynchus anatinus* (TOP 14 LEDGE) in Australia.

The loss-significant areas are mainly located in Amazonia, Central America and Central Africa (Fig. 4b). They mostly correspond to the hotspots of richness in TOP LEDGE species (Fig. 3b and Supplementary Table 1), and the loss in expected PD if all species of the cell became extinct is strongly correlated to the richness in TOP LEDGE species ($\rho = 0.85$, Supplementary Fig. 4).

**Conservation targets to maximise gains in expected PD differ from conservation targets to minimise losses in expected PD.** HEDGE and LEDGE scores are weakly correlated ($\rho = 0.24$, Supplementary Fig. 5), reflecting the difference in focus – TOP HEDGE species typically have high threat; TOP LEDGE species typically have low threat. Only 20% of the loss-significant species

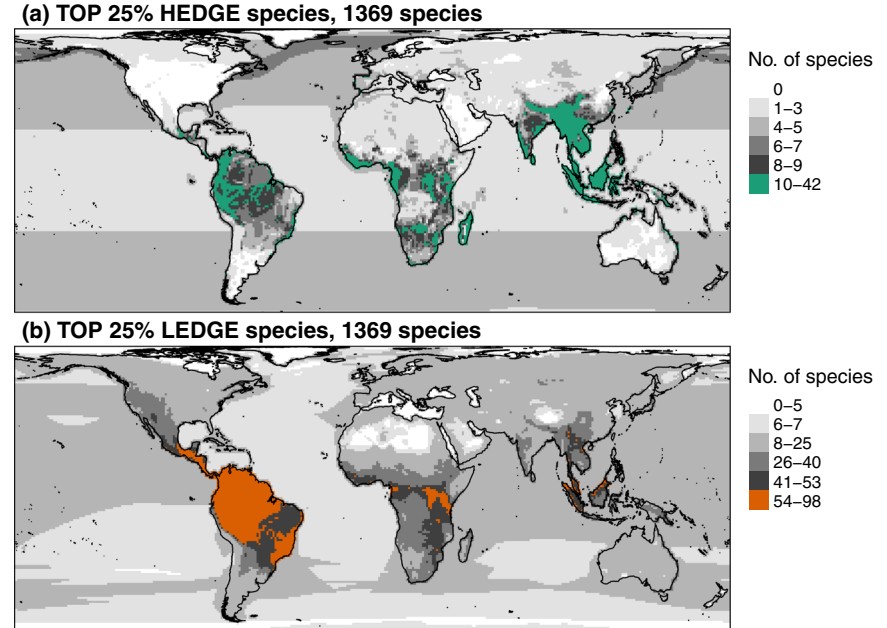

**Fig. 3 Spatial patterns and hotspots of species richness of the species of interest for conserving global mammalian PD.** Spatial patterns and hotspots of species richness of the species of interest for conserving global mammalian PD. Spatial patterns and hotspots of species richness for the TOP 25% HEDGE species (**a**) and the TOP 25% LEDGE species (**b**). Hotspots, in colour, represent the 2.5% richest cells.

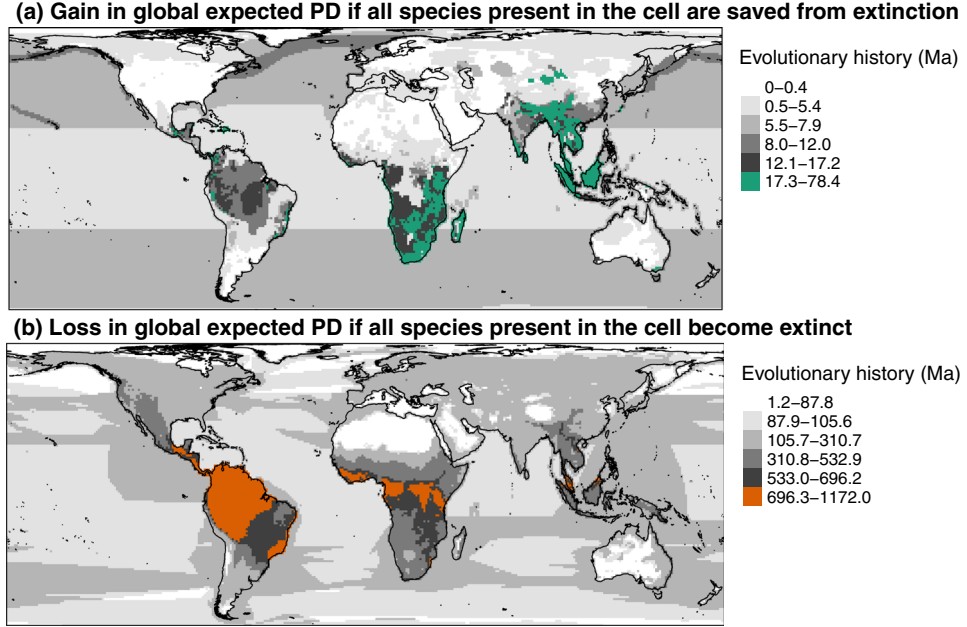

**Fig. 4 Areas of interest for conserving global mammalian PD.** Areas of interest for conserving global mammalian PD. Spatial patterns and hotspots of gain in global expected PD if all species present in the cell are saved from extinction ($p = 0$) (**a**) and loss in global expected PD if all species present in the cell become extinct ($p = 1$) (**b**). Hotspots, in colour, represent the 2.5% cells with the highest values.

are also priority species (Supplementary Table 2). This includes the aye-aye *Daubentonia madagascariensis* (TOP 19 LEDGE and TOP 6 HEDGE), the two *Solenodon* species *S. cubanus* and *S. paradoxus* (TOP 26 LEDGE and TOP 8 HEDGE *ex-aequo*) and the blunt-eared bat *Tomopeas ravus* (TOP 51 LEDGE and TOP 13 HEDGE). Overall, only evolutionarily very distinctive and moderately threatened species will belong to both TOP 25% HEDGE and TOP 25% LEDGE ranks.

Similarly, correlation between the gain in expected PD if all species of the cell were secured and the loss in expected PD if all

species of the cell became extinct is low ($\rho = 0.23$, Supplementary Fig. 4). The priority areas whose protection would bring the most important gains in expected PD generally do not match the loss-significant areas whose damage would trigger the most important losses in expected PD. The only 12% of grid cells in common are mainly located in Kenya, Malaysia, Columbia, Peru and Brazil (Fig. 4 and Supplementary Table 1).

These findings illustrate how conservation strategies to maximise gains or to minimise losses of expected PD not only differ from a conceptual point of view, but also may result in

distinct conservation targets. These two types of strategies nevertheless provide useful complementary targets. On the one hand, species whose protection would maximise gains in expected PD are very evolutionarily distinctive and currently threatened. Thus, targeting such priority species is urgent, as they are on the verge of extinction and offer positive opportunities for large conservation gains. But focusing only on them is very risky, as they require huge recovery efforts with unknown chances of success[43]. The same holds true for areas whose protection would maximise gains in expected PD, which concentrate evolutionarily distinctive and threatened species. Such priority areas therefore fit the definition of both a World Heritage Site, because each contains "threatened species of outstanding universal value from the point of view of science or conservation" (https://whc.unesco.org/en/criteria/) and a Phylogenetic Key Biodiversity Area, because each "makes a big contribution, as long as it persists, to averting expected PD loss"[7].

On the other hand, species whose protection would minimise losses in expected PD (i.e. those whose extinction would trigger the most important losses in expected PD) are highly evolutionarily distinctive and typically less threatened than priority species. They represent a different form of positive messaging for conservation – they are not threatened, but deserve our appreciation in securing a large amount of PD. Thus, although there is not a red alert indicating the imminent loss of these species, each new human action that might damage them should be avoided[16,44]. In addition, it is possible that these less threatened species might be further secured for relatively little cost[43]. The same is valid for areas whose protection would minimise losses in expected PD, which concentrate evolutionarily distinctive species less threatened than priority species. It means that human actions threatening mammals in the loss-significant areas identified here, such as the massive deforestation in Amazonia[45] or in tropical forests of sub-Saharan Africa[46], are very worrying as they may trigger a huge loss of global mammalian PD. Contrarily, human actions protecting mammals in these loss-significant areas have the power to secure important amounts of global mammalian PD for comparatively little effort.

**Priority species to maximise gains in expected PD overlap with previously identified priorities, but not priority areas**. HEDGE and EDGE scores are highly correlated ($\rho = 0.94$) and most of the priority species (87%) are also among the species with the highest EDGE score (TOP 25% EDGE; Supplementary Fig. 5 and Supplementary Table 2), i.e. the metric used to identify priority species by the EDGE of Existence Programme (www.edgeofexistence.org) – the only global conservation initiative to focus specifically on threatened species that represent a significant amount of unique phylogenetic diversity. Even though the high correspondence between HEDGE and EDGE was expected (as both give the highest scores to species that are very evolutionarily distinct and very threatened) and is consistent with previous findings[47], there are noteworthy differences. For instance, among the bottom half of the TOP 25% HEDGE species, some do not belong to the TOP 25% EDGE, such as the black-bearded flying fox *Pteropus melanopogon* (Supplementary Fig. 5). This species is among the least original but is endangered within a threatened lineage (34 out of the 58 species of the *Pteropus* lineage are threatened). Thus, despite the fact that it is not a priority EDGE species, managing to secure it would still bring more important gains in expected PD than securing any of the 4657 mammal species with a lowest HEDGE score.

With a percentage of grid cells in common ranging from 22 for species richness (SR, number of species) to 52 for threatened SR (number of threatened species), the newly identified priority areas generally weakly overlap with hotspots of current species-based or phylogeny-based scores – except those in Madagascar (Supplementary Fig. 6 and Supplementary Table 1). Specifically, only a few priority areas are located in Amazonia (they are rather concentrated in South-East Africa, Madagascar, and South-East and Central Asia), while Amazonia concentrates hotspots of SR, threatened SR, PD and threatened PD (i.e. PD in the set of threatened species). In other words, while protecting Amazonia is a good strategy to maintain high levels of SR and PD, maximising gains in expected PD would better be achieved by protecting South-East Africa, Madagascar, and South-East and Central Asia. This may be because these priority areas house more threatened lineages than Amazonia, but this would require further investigation. Correlations between gain in expected PD if all species of the cell were secured and species-based scores are generally low (Supplementary Fig. 4), except for threatened SR ($\rho = 0.86$). Likewise, correlations between gain in expected PD if all species of the cell were secured and phylogeny-based scores are generally low (Supplementary Fig. 4), except for threatened PD ($\rho = 0.64$). Therefore, and in contrast to the results for priority species, very few of the priority areas whose protection would maximise gains in expected PD overlap with biodiversity hotspots previously used to define priority areas for mammal conservation at a global scale[1,2,23,48]. Moreover, these newly identified priority areas hardly ever overlap with previously identified priority areas by spatial conservation planning, i.e. defined as the set of complementary areas that best represents global mammal diversity. This holds true for both previously identified priority areas best representing species diversity[49] and phylogenetic diversity[22] – even if some areas, such as parts of Madagascar and Indonesia, are priority areas according to our criteria and also belong to the set of areas best representing species and phylogenetic diversity. This low overlap between priority areas identified here and those previously identified means that the priority areas identified here have few chances to be detected by other approaches. While it has been previously argued that species data can be good surrogates of PD for spatial conservation planning when the aim is to maximise current extant PD[25], our work suggests this is not true when the aim is to maximise gains in expected PD.

**Current protection and guidance for future management of the newly identified species and areas of interest**. Sixty-eight per cent of the priority species (TOP 25% HEDGE) are protected by at least one conservation measure, and land/water management is the most common conservation measure for these priority species (Fig. 5a and Supplementary Data 2). However, 32% of the priority species do not benefit from any conservation measure, including the New Zealand greater short-tailed bat *Mystacina robusta*, which is the TOP 2 HEDGE species. These priority species therefore require urgent conservation attention.

The majority of the loss-significant species (TOP 25% LEDGE) do not benefit from any conservation measure (54%), including the aardvark *Orycteropus afer* (TOP 1 LEDGE); and for the species benefitting from at least one conservation measure, land/water management is the most common measure (Fig. 5b and Supplementary Data 3). Because these loss-significant species tend to be globally secure, conservation measures for these species are not urgent, but more preventive conservation measures would nonetheless help to secure important amounts of global mammalian PD.

The need of urgent conservation attention for some priority species also holds true for most priority areas as, overall, only 18.3% of the total surface of the priority areas (i.e. over all priority areas) is covered by the existing global network of protected areas.

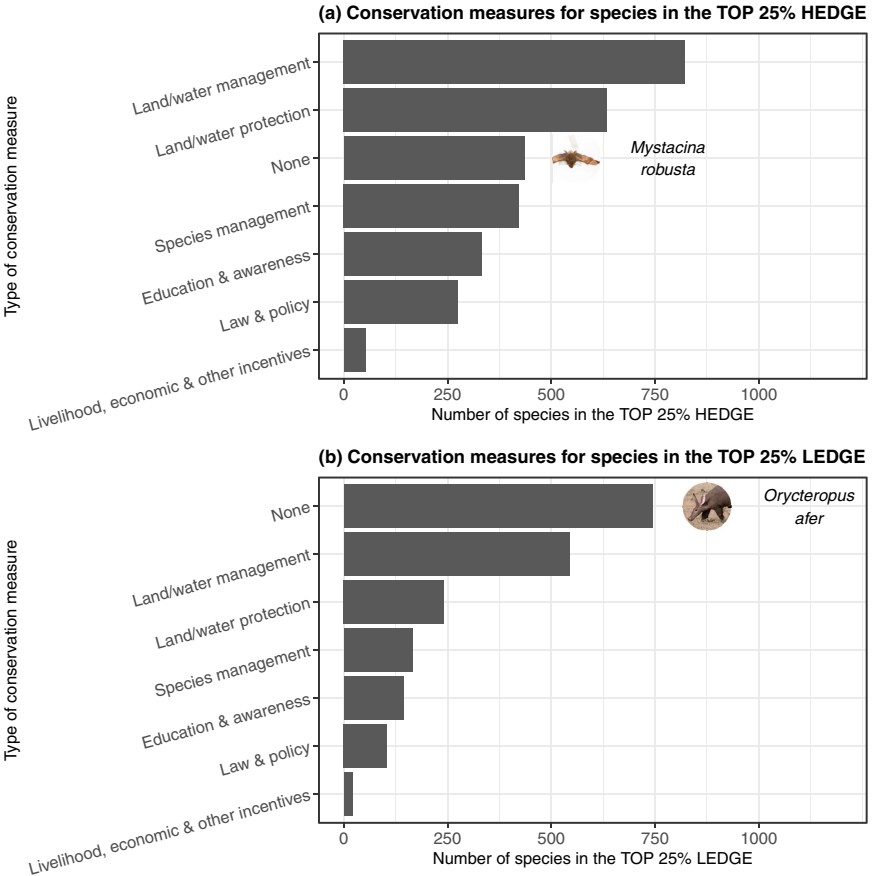

**Fig. 5 Conservation measures of the species of interest for conserving global mammalian PD.** Conservation measures for the TOP 25% HEDGE species (**a**) and the TOP 25% LEDGE species (**b**). The images illustrate examples of unprotected species. The image for *Mystacina robusta* has been modified from https://www.inaturalist.org/photos/13050710 (attribution: Auckland Museum, license: CC BY 4.0, some rights reserved). The image for *Orycteropus afer* has been modified from https://www.inaturalist.org/photos/78448701 (attribution: Dave Brown, license: CC0 1.0, no copyright). Source data are provided as a Source Data file.

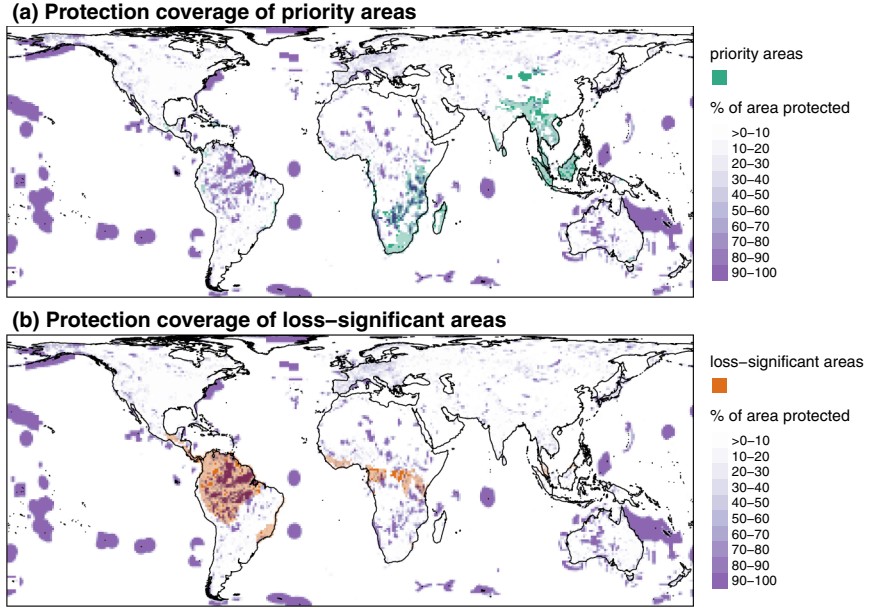

**Fig. 6 Protection coverage of the areas of interest for conserving global mammalian PD.** Spatial overlap between the percentage of area protected by the current network of protected areas and the priority areas (hotspots of gain in global expected PD if all species present in the cell are saved from extinction) (**a**) or the loss-significant areas (hotspots of loss in global expected PD if all species present in the cell become globally extinct) (**b**). The priority areas not protected appear in dark green, those slightly protected in light green, and those heavily protected in blue. The loss-significant areas not protected appear in dark orange, those slightly protected in light orange, and those heavily protected in plum.

The protection coverage of the priority areas (i.e. within each priority area) ranges from 0% to 99.9%. On one hand, 193 priority areas, representing 15% of all priority areas, are not covered at all by the global network of protected areas (Fig. 6a). On the other hand, 147 priority areas, representing 11.5% of the priority areas, have at least half of their surface under protection. The protection coverage of the priority areas varies according to their main locations: it is higher in the northern parts of South-East Africa than in South Africa, Madagascar and in Asia (Fig. 6a). Here we highlight where the priority areas lack protection coverage (mainly in South Africa, Madagascar and in Asia) and this can guide the decision-making process regarding where to add new, strictly protected areas to increase gains in expected PD.

Interestingly, loss-significant areas are better protected than priority areas, as 32.5% of the total surface of the loss-significant areas (i.e. over all loss-significant areas) is covered by the existing global network of protected areas. Nonetheless, we stress out that some loss-significant areas are poorly protected, notably in some parts of Amazonia and Central Africa (Fig. 6b). Loss-significant areas in these regions are good candidates to add new, non-necessarily strictly protected areas or to take other area-based conservation measures to prevent losses in expected PD. Overall, our findings echo previous ones showing that the existing network of protected areas is inadequate to protect PD in different vertebrate groups[23,24,50,51]. This drawback could be overcome by explicitly including PD targets in biodiversity policy frameworks.

**Limits and perspectives**. Here, we defined conservation priorities with one clear objective: maximising gain in expected PD for mammals globally. For this reason, we considered mammals from the three different types of biomes (i.e. terrestrial, freshwater and marine) together – despite the fact that conservation actions and agendas are separate for the continental (i.e. terrestrial and freshwater) and the marine biomes. Consequently, as there are less mammal species in the sea than on land, the marine environment is therefore not represented in our priority areas. Nevertheless, 22% of marine mammals are priority species – including all Sirenians – and priority species represent a high proportion of total species richness in the northern parts of the Atlantic and Pacific Oceans and in the Caspian Sea. In addition, there would be probably more marine priority areas if the objective was to maximise gain in expected PD over multiple taxonomic groups globally, including groups that are mostly marine. As phylogenetically informed priorities have been defined for mostly marine groups such as corals[41] and sharks and rays[52], our approach to identify priority areas could be easily extended to additional taxonomic groups better representing all environments in the near future.

Further, the conservation priorities identified here represent the set of species or areas whose individual protection would maximise our PD metric (gain in global expected PD). Thus, it differs from alternative phylogenetically informed priority settings that take into account the complementarity between species[7,17,53] or areas[17,18] so that the priorities represent the optimum set of species or areas that collectively maximise PD. We note that in practice, conservation strategies often depart from the idealised set of species or areas that collectively maximise PD[22]. Our conservation priorities for individual species (or areas) can be recalculated to respond to ongoing changes in the threat status of species. In this way, our work can support further development of phylogenetically informed spatial conservation planning[24–26] to identify the optimum set of areas that would collectively maximise gain in expected global PD.

Moreover, we had to down-sample the WDPA dataset to analyse the protection coverage of priority areas at the resolution they were identified (i.e. 96.5 km × 96.5 at 30° North and South), which caused the loss of spatial information. While this approach is useful to highlight the priority areas that completely lack protection coverage, it is not sufficient to analyse the quality of protection in priority areas that are partially protected. Priority areas with a low percentage of surface protected may actually be well protected if they contain small, numerous, well-connected and well-managed protected areas. Contrarily, priority areas with a high percentage of surface protected may not be well protected if they contain only one large protected area, not connected to others and not well managed. Therefore, the quality of protection in each priority areas need to be further investigated at finer scale, for instance by using the IUCN categories for protected areas.

Finally, we used IUCN designations projected to 50 years to derive probabilities of extinction from Red List extinction risk categories[54] for each species and calculate expected PD in 2070. It means that our baseline scenario for 2070 corresponds to the remaining PD we would observe if, from now on, nothing could change species trajectories to extinction. The species and areas of interest we defined correspond to those whose protection would maximise gains (priority species/areas) or minimise losses (loss-significant species/areas) in expected PD compared to this baseline scenario (Fig. 1). Our results are dependent on the model we chose to calculate species' probabilities of extinction. In particular it does not consider how conservation actions could alter these probabilities in the next decades. Further studies would be needed to investigate which existing model of species extinction is most suitable for concrete conservation actions – as, by definition, the HEDGE, LEDGE and other expected PD statistics are dependent on the definition of species' extinction probabilities. Further studies are also needed to improve current estimations of these probabilities. As new models are being developed to estimate probabilities of extinction[55], we note that our approach could be applied to any baseline scenario using such new models.

**Concluding remarks**. Both the increasing recognition that preserving PD is a sound mean to maintain biodiversity option values for humanity[5,6,18,30,56–58] and the growth of the EDGE of Existence Programme highlight the importance of biodiversity policies and conservation actions that specifically target PD. The strategy we used here to identify priorities for conserving global mammalian PD directly reflects the opportunity to increase expected PD, and its robustness to recent phylogenetic updates suggests that these priorities will remain the same even with new phylogenetic knowledge. The strategy we used to identify loss-significant species and areas that would limit losses in expected PD is only moderately robust to recent phylogenetic updates, indicating that the identity of these loss-significant species and areas may need to be updated with new phylogenetic knowledge. Nonetheless, implementing those strategies in multiple taxonomic groups, i.e. maximising gains and minimising losses of expected PD, is a proactive way to reach a conservation target of net positive outcomes for PD. Because it reflects the positive effects conservation actions can have on expected PD, such a conservation target offers a more positive vision for the future than imagining the consequences of further PD losses only. This may therefore help to widen the engagement of decision-makers in PD conservation. We thus hope that our work will, together with other initiatives such as the IUCN Species Survival Commission Phylogenetic Diversity Task Force, inform the CBD's Scientific Subsidiary Body and encourage the inclusion of a

conservation target requiring net positive outcomes for PD in the CBD post-2020 framework.

## Methods

**Phylogenies, range maps and threat status**. We obtained phylogenies, range maps and threat status for all 5477 extant mammal species from the taxonomically integrated platform PHYLACINE[37,38]. For phylogenies, we extracted 100 trees from the available posterior distribution of 1000 trees to take into account phylogenetic uncertainties. Range maps correspond to the projection of IUCN polygon range maps of the current and natural range for each species[59] into a Behrman cylindrical equal area raster with a resolution of 96.5 km by 96.5 km at 30° North and 30° South and plotted between 90°N and 60°S. Threat status rankings were taken from IUCN Version 2016-3[59]. For the 777 data-deficient species, we imputed their threat status based on their range size following Véron et al.[60].

**Calculation of species scores and identification of species of interest**. We computed four phylogeny-based scores for each of the 5477 extant mammal species. We first calculated evolutionary distinctiveness (ED)[13,61] using the function *distinctTree* from the R package 'adiv'[62,63]. Second, we computed evolutionary distinctiveness and global endangerment (EDGE)[13] expressed as follows:

$$EDGE = \ln(1 + ED) + GE \times \ln(2), \quad (1)$$

where GE is the Red List category weight [least concern = 0, near threatened = 1, vulnerable = 2, endangered = 3, critically endangered = 4]. Third, we calculated heightened evolutionary distinctiveness and global endangerment (HEDGE) – the φi' score in Steel et al.[16] – corresponding for each species to the gain in expected PD if this species was secured (probability of extinction goes to 0). Finally, we computed a score corresponding for each species to the loss in expected PD if this species went extinct (probability of extinction goes to 1) – the φi" score in Steel et al.[16]. We will refer to this as the 'loss-significant evolutionarily distinctive globally enduring' score (LEDGE); while the LEDGE concept focuses only on secure species[20], we use the term here to describe the calculation applied to any species. HEDGE and LEDGE scores were computed using the self-written function *HED2* (see R script "new_functions" in the GitHub repository[64]), with the IUCN designations projected to 50 years to derive probabilities of extinction from Red List categories[54]. For all scores, we repeated the calculations over the 100 trees and computed the median species score over the 100 calculations. We tested the correlation between each pair of scores using a Spearman correlation test.

For each score, we identified the 25% of species having the highest scores following Stein et al.[52]. Specifically, using our conceptual framework to identify species of interest for conserving global mammalian PD (Fig. 1), we identified priority species as the 25% of species having the highest HEDGE scores: these species are the typically currently threatened, distinctive species whose protection would mean a big gain in expected PD, regardless of their conservation status. We also identified loss-significant species as the 25% of species having the highest LEDGE scores: these species are the typically currently secure species whose loss would mean a big loss in expected PD regardless of their conservation status because they are currently distinctive or because they have expected future distinctiveness due to the high extinction risk of their close relatives. We compared the 25% of species having the highest scores for each pair of scores.

To test if our results were robust to recent updates of the mammalian phylogeny, we replicated our calculations of HEDGE and LEDGE scores using the phylogeny of Upham et al.[39]. We then tested the correlation between scores calculated with the phylogeny from PHYLACINE[37,38] and those calculated with the phylogeny of Upham et al.[39] using a Spearman correlation test. We also compared the priority species and the loss-significant species identified with the two phylogenies.

**Conservation measures of species of interest**. We obtained the list of conservation measures for the TOP 25% HEDGE species and the TOP 25% LEDGE species from IUCN Version 2019-2[65]. As taxonomy in IUCN Version 2019-2 has changed since IUCN Version 2016-3, we first updated species names from IUCN Version 2016-3 to IUCN Version 2019-2 using the function *rl_synonyms* from the R package 'rredlist'[66]. For nine remaining problematic names, we made manual edits (see R script "figures_tables" in the GitHub repository[64] for details) based on the taxonomic information in IUCN Version 2019-2[65] and in the third version of Mammal Species of the World[67]. We obtained the list of conservation measures using the function *rl_measures* from the R package 'rredlist'[66] and only kept their first level of classification with 6 categories: "Land/water protection", "Land/water management", "Species management", "Education & awareness", "Law & policy", and "Livelihood, economic & other incentives".

**Calculation of scores by grid cell and identification of areas of interest**. We overlaid the PHYLACINE gridded range maps for each species[37] to record their presence and absence in the 51,120 cells of a reference gridded map having, by construction, the same spatial extent, resolution and projection than each individual gridded range map. On the basis of these gridded species range maps, we calculated 12 scores. We first computed four species-based scores:

(i) species richness (SR), as the total number of species;
(ii) threatened species richness (TSR), as the total number of threatened species (i.e. vulnerable, endangered and critically endangered species);
(iii) rare species richness (RSR), as the total number of rare species (here defined as species having a range size below the median range size of the 5477 extant mammal species);
(iv) species-weighted rarity (SWR), corresponding to the sum of species weights in a grid cell where each species weight is the inverse of the number of grid cells where the species occurs, and permitting to identify areas with high number of spatially-restricted species.

We then computed the phylogeny-based equivalents of these four species-based scores:

(v) phylogenetic diversity (PD)[5], using the function pd from the R package 'picante'[68];
(vi) threatened phylogenetic diversity (TPD), corresponding to PD of threatened species only;
(vii) rare phylogenetic diversity (RPD), corresponding to PD of rare species only;
(viii) phylogenetic-weighted rarity (PWR), previously introduced by Rosauer et al.[69] as phylogenetic endemism and permitting to identify areas with a high concentration of spatially-restricted PD, here calculated by updating the function calc_PE (original function: https://github.com/DanRosauer/phylospatial/tree/master/PhyloEndemism_in_R; see the R script "calc_PE" in the Github repository64 for the updated function).

We also computed:

(ix) the number of species in the TOP 25% HEDGE to identify the areas with a high concentration of priority species;
(x) the number of species in the TOP 25% LEDGE to identify the areas with a high concentration of loss-significant species.

Finally, we computed two new scores that represent respectively best and worst scenarios of expected PD in the next 50 years (Fig. 1):

(xi) the gain in expected PD if all species present in a grid cell were secured (GexpPD), corresponding to the gain in expected PD if a local conservation action in the grid cell was able to transform the current global probability of extinction of all species present in the cell to 0;
(xii) the loss in expected PD if all species present in a grid cell became extinct (LexpPD), corresponding to the loss in expected PD if a local threat in the cell was able to transform the current global probability of extinction of all species present in the cell to 1.

Although the scenario underpinning the calculations of GexpPD (resp. LexpPD) is unlikely to happen, it has the advantage of representing the best (resp. worst) outcome we can expect from the human protection (resp. threat) of a local area: securing the global persistence (resp. triggering the global disappearance) of all species present in the locally protected (resp. threatened) area. Moreover, areas with high GexpPD scores may correspond to phylogenetic key biodiversity areas (PDKBAs) since they make a big contribution, as long as they persist, to averting expected PD loss[7,20].

For the phylogeny-based scores, we repeated the calculations over the 100 trees and computed the median cell score over the 100 calculations. We tested the correlation between each pair of scores using a Spearman correlation test and we defined hotspots for each score as the 2.5% grid cells with the highest values[1,70]. Specifically, we identified priority areas as the hotspots of gain in expected PD if all species present in a cell were secured, regardless of their conservation status (GexpPD): they represent the areas where protection would mean important benefits in terms of PD. We also identified loss-significant areas as the hotspots of loss in expected PD if all species present in a cell became extinct, regardless of their conservation status (LexpPD): they represent the areas where damage would mean important losses in terms of PD. We compared the 2.5% of grid cells with the highest scores for each pair of scores.

In addition to the 12 scores described above, we also mapped:

the proportion of threatened species (TSR/SR), of rare species (RSR/SR) and mean species-weighted rarity (SWR/SR) to represent the areas exhibiting high proportions of rare and threatened species;
the proportion of threatened PD (TPD/PD), of rare PD (RPD/PD) and mean phylogenetic-weighted rarity (PWR/PD) to represent the areas exhibiting high proportions of threatened and rare PD;
the proportion of TOP 25% HEDGE species and of TOP 25% LEDGE species to highlight the areas exhibiting a high proportion of priority and loss-significant species.

**Matching the global network of protected areas and the priority areas**. We downloaded the public version of the World Database on Protected Areas (WDPA) for November 2019 from Protected Planet (https://www.protectedplanet.net/). The WDPA is managed by the World Conservation Monitoring Centre (WCMC) of the United Nations Environment Program (UNEP) in collaboration with the IUCN[71]. We excluded protected areas (i) whose status was "Not Reported" or "Proposed", (ii) designed as "UNESCO-MAB Biosphere Reserve", and (iii) provided as points with unknown boundaries and without a reported area. However, we included protected areas provided as points with unknown boundaries but with a reported

area, using a geodesic circular buffer with an area equal to the reported value. We solved problems of invalid geometries in the selected files, and then dissolved all polygons – removing this way overlaps and redundancy within protected areas. We then transformed this re-processed WDPA dataset to fit the extent and projection of the reference gridded map. Finally, we calculated the percentage of area covered by the re-processed and transformed WDPA dataset for each grid cell of the reference gridded map. We carried out all these steps with a combination of open-source tools (GDAL 3.0, GRASS GIS 7.8, PostgreSQL 12/PostGIS 3.0). We overlaid this map of percentage of area protected with the map of priority areas to derive the percentage of area protected for each of the priority areas identified.

**Reporting summary**. Further information on research design is available in the Nature Research Reporting Summary linked to this article.

## Data availability
This study used datasets that are publicly available from PHYLACINE (https://doi.org/10.5061/dryad.bp26v20) and VertLife (https://data.vertlife.org/). It also uses data from IUCN version 2019-2 that are available from the corresponding author upon request and with permission of IUCN. The coastline dataset used to make Figs. 3, 4, 6, and Supplementary Figs. 2, 6 and 7 is available from Natural Earth (https://www.naturalearthdata.com/downloads/50m-physical-vectors/). All datasets generated during this study are publicly available on the following GitHub repository[64]: https://github.com/MarineRobuchon/consmampd. Source data are provided with this paper.

## Code availability
The code used for the analyses is publicly available on the following GitHub repository[64]: https://github.com/MarineRobuchon/consmampd.

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

## Acknowledgements

This work was supported by the French State through the Research National Agency under the LabEx BCDiv [ANR-10-LABX-0003-BCDiv], within the framework of the program 'Investing for the future' [ANR-11-IDEX-0004-02], and by the institutional activities of the Global Observatory for Biodiversity and Ecosystem Services (GLOBES) project of the European Commission.

## Author contributions

M.R., S.P. and B.L. designed the study with inputs from D.F., R.P. and S.V.; M.R., B.L., A.M., G.De and S.V. performed the analyses. G.Du improved the manuscript in terms of policy relevance. M.R. coordinated the study and wrote the paper with inputs from all the other authors (S.P., S.V., G.De, D.F., A.M., R.P., G.Du and B.L.).

## Competing interests

The authors declare no competing interests.
