## [Peer Review File · Nature Communications]

REVIEWER COMMENTS

Reviewer #1 (Remarks to the Author):

The manuscript is very interesting and well written. I enjoyed reading and it raised some comments/questions for me:

I would change the “revisiting” by “Identifying” (or something similar) in the title because I think it gives the impression that the ms will directly compare the priorities identified here with existing others.

I believe that more explanation is needed to separate your species/area priorities from existing others. There are plenty of papers regarding PD conservation. How does the current manuscript stand out from them? How can HEDGE/LEDGE inform priorities in a way the others can not? It is still hard for me to understand how they can address virtually the same issue and present different results.

I feel that more arguments on “why should we care about PD” in the introduction. I believe the paper from Tucker et al. (2019, Assessing the utility of conserving evolutionary history) might help. Also, It is already a struggle to convince governments and other decision-makers into the need to protect taxonomic diversity, How can we engage stakeholders and decision-makers into PD conservation?

l111. “...both biomes..” Do you mean terrestrial and aquatic?

Considering that conservation actions (policies, management, etc) are taken at national/regional levels, and the discussion for terrestrial and marine conservation agenda is separate, do you feel it is adequate to analyze them together?

I missed some more discussion among the results. For example, it seems, by your results, that Australia does not stand out in any of the results. It is surprising considering the evolutionary distinctiveness and threat of its mammalian diversity. Why is that?

l273 "This low overlap between priority areas identified here and those previously-identified means that the priority areas identified here have few chances to be detected by other approaches"

That sentence made me very intrigued, because you did not actually overlap your priorities with the others and, as far as I remember, I am not sure if they are largely different from yours. Also, What precisely your framework is getting that others are missing? Make that clear.

Reviewer #2 (Remarks to the Author):

1-What are the major claims of the paper?

Robuchon et al. aimed to inform global mammalian conservation based on the gain in expected PD if priority species are protected. This was done using HEDGE and LEDGE scores. They showed that priority species and areas for conservation not only do not match the already-identified species and areas based on EDGE scores but are not well captured within the existing network of protected areas. The rationale for the study can be summarized in 4 points:

- There is no information on HEDGE and LEDGE scores for global mammalian diversity (L107-108).
- Although several studies have identified PD-based priority areas for mammals, none of them considered simultaneously both terrestrial and aquatic mammals,
- Of these studies, only one regional study has applied the expected PD framework of the HEDGE and LEDGE scores to identify priority sites (L109-113).
- Finally, a strategy requiring net outcome for PD would encourage wider engagement in PD-driven nature conservation (L114-119).

2-Are they novel and will they be of interest to others in the community and the wider field?

Setting priority for conservation is not novel; however, the approach used in the study is different from most studies: As opposed to most prioritisation strategies that seek to avoid further biodiversity losses, Robuchon et al. focused on expected gain of PD if priority species and areas are protected. In Line 114-119, they claim that their approach would encourage wider engagement in PD-driven nature conservation. The problem with this claim is that since 1992 that PD has been proposed, it hasn't been adopted widely in on-the-ground conservation actions and even in biodiversity policies as the authors themselves rightfully reported in L76-83. Does this suggest that decision makers do not believe in the added value of PD in conservation? One may think so (I do not share that view). So, how can one be sure that HEDGE approach here would avert this, i.e. convincing decision makers to act otherwise? Some even argue that PD seems to be more academic exercises than anything else. I suggest that the authors reword their claim that the use of HEDGE and LEDGE would encourage wider engagement in PD conservation.

3-If the conclusions are not original, it would be helpful if you could provide relevant references.

As expected, they reported a high correlation between HEDGE and EDGE scores ($\rho = 0.94$) leading to the finding that 87% of HEDGE priority species are also EDGE priority species (L244-245). Interestingly, priority areas based on HEDGE overlap poorly with traditional biodiversity hotspots (L260-269). The authors claim that this low overlap is the good reason why HEDGE based conservation should be embraced in international biodiversity policies (L275-279). This argument needs to be strengthened. Conservation resources are already limited, traditional conservation measures based on SR are not always well implemented due to pressing development priorities in most species-rich countries, the PD since 1992 is not yet embraced in on-the-ground conservations nor in biodiversity policy, etc. That's why

the argument consisting of saying the newly delimited priority species or priority species need to be embraced since they are different from the traditional ones doesn't seem to be convincing or exciting to me, although I fully agree that we need to be calling for PD to be included in biodiversity policy and actions. I am expecting more practical conservation implications than this, e.g. do we need to expand network of PA or Do we need to re-design the network to include the new priority areas and species? etc.

4-Is the work convincing, and if not, what further evidence would be required to strengthen the conclusions?

The work is convincing as findings are well supported by the analysis of data collected, except that I do not get the rationale of the authors when they claim that the use of HEDGE will "widen the engagement of decision-makers in conservation".

5-On a more subjective note, do you feel that the paper will influence thinking in the field?

I have no doubt that the paper will draw attention of conservation biologists especially in the academic environment.

6-Please feel free to raise any further questions and concerns about the paper.

L129 In contrast to previous works (plural for work)

L162 Fig 3a shows west, central, east and southern Africa and not just South Africa.

L241mammalian PD for comparatively little effort.

L250of the TOP 25% HEDGE species, some... (comma after species)

L259-260 What is the implication for conservation?

L293. not clear to me. In the sentence above (L291-92), you said only 18.3% of priority areas is covered by the network but here you say the protection coverage is 0-99.9%. confusing.

L302-303: I agree fully.

L305: Figure 6. This spatial overlap should be done and analyzed for both HEDGE and LEDGE. It's important to see the extent of coverage of LEDGE as well in the existing network of PAs.

L310-327 I do not see the EU recommendation to eradicate this *M. coypus* (or any other alien invasive species) as a conservation paradox, in light of your recommendations to avoid human actions that might damage this species. I think your recommendation to avoid human actions on priority species should be more focused on priority species in their native range i.e. new human actions that may damage species in their native range should be avoided. Alien invasive species are themselves threatening biodiversity in their recipient environment. By avoiding any intervention to eradicate or control alien invasive species in their recipient environment, the risk of losing local diversity is high. I suggest that conservation measures of problematic species like alien should focus on native ranges. Or when in non-native range, they should be in controlled environment, e.g. Zoos.

L347-350 I agree fully with this type of argument.

L364: EDGE (not Edge?)

370-371 "...has the power to widen the engagement of decision-makers in PD conservation". I do not buy this argument as I indicated above. The integration of evolution in conservation is lacking visibly buy-in from decision- or policy-makers. I don't see how the use of HEDGE, as promoted in this study, will change this.

7- We would also be grateful if you could comment on the appropriateness and validity of any statistical analysis, as well the ability of a researcher to reproduce the work, given the level of detail provided.

The data analysis including the statistics is great, and I can't see any flaw on this regards. The R script used is not publicly available yet but will be after the paper is published.

Reviewer #3 (Remarks to the Author):

Referee's report for Manuscript 266789 (Robuchon et al.)

The core of this study is a consideration of which species of mammal (and in aggregate, which places where they occur) whose conservation (their prob(extinction) is set to 0) would safeguard most "expected" (aka projected future) phylogenetic diversity, or whose loss (their prob(extinction) set to 1) would lead to the greatest loss of expected phylogenetic diversity. The ingredients for this exercise are a dated phylogenetic tree of all mammals (from Phylacine 2.0), a Red List index from the IUCN converted to a p(ext), and a map of the species extents of occurrence.

There is a great deal to commend in this work, and it represents a large investment in time and thought. I think the conceptual switch presented in figure 1, where one considers species from the contrasting points of view of how they contribute to future diversity (the HEDGE metric), or how their loss might contribute to future loss of diversity (the LEDGE metric) is elegant.

That said, I have two concerns with the paper at a high organizational and data-stream level, neither likely to be welcome to the authors. The set-up of this paper is decidedly political – the authors are explicit that this exercise is in aid of convincing policymakers to consider PD in the policy being created for the revision of the Convention on Biological Diversity in 2021. While this is an admirable (if perhaps still premature) goal (at least till some new papers see the light of day), it does mean one has to particularly conservative.

I start with the easier or the two issues. The Phylacine 2.0 tree produces fairly different DR scores than do point estimates from the newer and better-dated Upman et al. (2019) trees of mammals (see figure 6(b), middle panel from that paper, which compares these numbers directly). $DR=1/ES$, and ES and ED

are generally strongly positively correlated (in my experience, with $\rho \gg 0.7$); both are strongly dependent on the pendant edge length, and that is the edge that also contributes most (by quite a ways) to HEDGE and LEDGE calculations. I think we need to know how sensitive the results in this paper are to the particular tree used: correlations across common species for the two (sets of) trees would be welcome, and at the very least, we need to know if the top 100 species are the same.

The second issue is thornier. A recent paper based on modelling work by Folmer Bokma (Monroe et al., Biol. Lett. 2019) estimated the instantaneous $p(\text{ext})$ of all birds over the past 30 years using Red List status changes, probabilities that can be projected over any time period (eg. 50 years). These data-based estimates are quite different from those derived from Red List definitions. For instance, their $p(\text{ext})$ over 10 years for a species designated as CR seems to be less than a tenth of the Red List definition: 0.04 at the upper limit (or 4%, $= 1 - (1 - 0.0039)^{10}$, taking the values from their supplementary table 2, which is supposed to ignore active downlisting, vs. the value of 0.5 pegged by the IUCN and used here). The corresponding Monroe et al. value for EN is less than 1/3 the Red List definition, so the differences are not due to a simple multiplier. This is because the Red List definitions are based on "no intervention" while the actual transition rates are estimated from observed changes that incorporate all that is done and not done for species on the list. Conservation attention changes a species' outlook, and this outlook changes with status, both which make sense. However, this suggests that forward projections based on the definitions are just plain wrong, because the projections are based on the assumption that we have, as of the time of writing, stopped all conservation activity. The authors of the Monroe et al. paper do not comment on this anywhere and seem unaware. Regardless, to the extent my comparison here is correct, projections created from Red List definitions (e.g. from Mooers et al. 2008 or the updated ones from Davis et al. PNAS 2018) are suspect.

Of course, the main results here (the species rankings) might be robust to the distribution of $p(\text{ext})$ used. Documenting that would increase the strength of the argument that HEDGE and LEDGE are powerful metrics of diversity protection. But until we know this, I do not advocate using expected PD in a policy-facing framework with Red-List-derived $p(\text{ext})$, as the conclusions might be misleading.

In the interests of time, I will stop my report here, in the hopes of getting a second version – if not at this journal, somewhere else. I have made a number of suggestions to the first part of the paper in MS Word and have included that as an attachment to this review. Overall, I think the argument that a HEDGE perspective is somehow a more useful "net outcome" might be compelling, but needs to be presented more clearly and in one place – currently it is spread across several different paragraphs. Further, I think the all-important discussion of expected PD, EDGE, HEDGE and LEDGE could use some work. "expected PD" needs to be introduced first (perhaps as "projected future PD").

So, for example, as an unrequested potential edit:

Global PD-based priority mammal species have been identified^{8,9} using the EDGE ('evolutionarily distinct and globally endangered') approach to attribute a priority score to a species by combining its

score of evolutionary distinctiveness (ED) with its status on the IUCN Red List of Threatened Species TM as an estimate of its extinction probability. The EDGE score is a crude projection of how much evolutionary history is "expected" to be lost, species by species, if no action is taken, based on the assigned probability of extinction of that species. This framework can be extended to the entire tree: if every tip is given an extinction probability (e.g. over the next 50 years), one can project how much of the tree will persist into the future, or the "expected PD" of the tree^{12,13}. One can in turn assign values to individual species, either the contribution to this expected PD of a species if it is secured (its probability of extinction is set to 0), or the loss of expected PD if it were to go extinct (its probability of extinction is set to 1)¹¹. We refer to the first projection as a HEDGE¹¹ score, and re-christen the latter as a LEDGE¹⁵ score. We note that the globally important EDGE of Existence Programme^{7,8} is endorsing expected PD calculations¹³ as an alternative to its conventional EDGE scores¹⁴.

HEDGE scores have tentatively been assigned to aquatic mammals¹⁰ but here we generate HEDGE and LEDGE values across all terrestrial and marine mammals globally, compare them, identify species with the highest values, and map species globally to identify hotspots. We find XXXX...

POINT-BY-POINT RESPONSE TO REVIEWER COMMENTS

Important note: the line numbering corresponds to the revised version of the manuscript showing all track changes (as requested by the editor), but the text copied/pasted within this response to reviewers corresponds to the revised clean version (not showing changes). We hope these clarifications will make the revision process more easy-going.

Reviewer #1 (Remarks to the Author):

R1.1. The manuscript is very interesting and well written. I enjoyed reading and it raised some comments/questions for me:

Reply to R1.1. We thank the reviewer for this very encouraging first comment! We answered all the comments and questions raised below.

R1.2. I would change the “revisiting” by “Identifying” (or something similar) in the title because I think it gives the impression that the ms will directly compare the priorities identified here with existing others.

Reply to R1.2. We actually compare the priorities here with existing others (lines 397-443, overlap results are shown in Table S3) so “revisiting” seems appropriate.

R1.3. I believe that more explanation is needed to separate your species/area priorities from existing others. There are plenty of papers regarding PD conservation. How does the current manuscript stand out from them? How can HEDGE/LEDGE inform priorities in a way the others can not? It is still hard for me to understand how they can address virtually the same issue and present different results.

Reply to R1.3. The priorities defined here correspond to the species/areas whose protection would bring the most important **gains in expected PD in a 50 years’ time horizon**.

For priority species, it corresponds to the species with the highest HEDGE scores. They already have been calculated for aquatic mammals, but not globally for all mammals.

For priority areas, it corresponds to the grid cells of the map with the highest gain in expected PD if all species present in the grid cell were globally secured. This has never been calculated globally for any group.

Contrary to alternative PD conservation frameworks that focus on maximising **CURRENT EXTANT PD**, the priorities we defined here focus on maximising **EXPECTED (ie projected in the future) PD**. This is why our priorities may differ from previously-defined PD priorities. We re-wrote the introduction to make this point clearer (lines 215-218):

“In contrast with existing PD conservation approaches that focus on maximising *current extant PD*, the species and areas of interest that we define here focus on maximising gains (priority species/areas) and minimising losses (loss-significant species/areas) of *expected PD in a 50 years’ time horizon* (Fig. 1)”.

In addition, we designed Figure 1 specifically to explain our framework and how it conceptually differs from other PD conservation frameworks.

R1.4. I feel that more arguments on “why should we care about PD” in the introduction. I believe the paper from Tucker et al. (2019, Assessing the utility of conserving evolutionary history) might help. Also, It is already a struggle to convince governments and other decision-makers into the need to protect taxonomic diversity, How can we engage stakeholders and decision-makers into PD conservation?

Reply to R1.4. We added more arguments on “why should we care about PD” in the introduction, including the reference of Tucker et al. 2019. However, we note that Tucker et al, while useful, follow community ecology traditions in using a broader notion of “phylogenetic diversity” and of “option value”. We therefore complemented that reference with the reference to Faith 2018, which explains some of these differences in terminology and how they affect how we talk about PD value. We also refer specifically to “biodiversity option value” to avoid confusion with the broader economics usage of “option value”.

This part now reads as follows (lines 102-11):

“Maximising PD is a sound conservation strategy as it maximises biodiversity at the level of features, which represents a reservoir of unanticipated future benefits for humanity (“biodiversity option values”)⁵. In accord with this, the Intergovernmental Platform on Biodiversity and Ecosystem Services (IPBES) in its global assessment report on biodiversity and ecosystem services⁶ adopted PD as an indicator of maintenance of options. Maximising PD may also preserve greater ecosystem functioning⁷ and increase the evolutionary potential of a species set so that one or more members of the set can adapt to changing conditions^{5,8}, although these issues are still debated and require more empirical evaluations^{9,10}. Beyond these utilitarian arguments, maximising PD is also a way to preserve the evolutionary heritage of our planet for its own value.”

Regarding the engagement of stakeholders in biodiversity/PD conservation: we explicitly build on the increasing appreciation that PD conservation benefits society by maintaining its options (“biodiversity option values”). We then build on this foundation by proposing a compelling conservation strategy requiring net positive outcomes for biodiversity (biodiversity gains > biodiversity losses) that should contribute to widen the engagement of stakeholders in biodiversity conservation, as argued by Bull et al. 2019: “A strategy requiring net positive outcomes — above and beyond targets for preventing further declines— would encourage wider engagement in nature conservation”. In this paper, we applied this proposal to PD, as explained in lines 181-207:

“Despite the increasing awareness that PD conservation would benefit society by maintaining its options and the recognition that it should be explicitly included into global conservation goals²⁸, the preservation of PD is not yet implemented in international biodiversity policies - such as the strategy to 2020 of the UN Convention on Biological Diversity (CBD)²⁹ and the new EU biodiversity strategy for 2030³⁰. These strategies include evolution explicitly only via the preservation of genetic diversity for cultivated plants, and farmed and domesticated animals. A few months away from the 15th Conference Of the Parties of the CBD which will set the new conservation targets to 2030, the first drafts of the post-2020 biodiversity strategy³¹ overlook genetic diversity³² and do not yet incorporate PD. A recent proposal to track two PD indicators in the CBD post-2020 framework¹⁶ includes the IPBES existing indicator for expected PD loss, but does not suggest any indicator that would highlight expected PD gains. To fill this gap and contribute to promote PD conservation within the post-2020 framework, we propose to adapt Bull et al.³³'s suggestion to move beyond strategies seeking to avoid further biodiversity losses and develop strategies resulting in net positive outcomes for biodiversity to

the particular case of PD. Indeed, strategies targeting net positive outcomes for biodiversity – i.e. biodiversity gains superior to biodiversity losses – would encourage wider engagement in biodiversity conservation³³, by highlighting the positive effects conservation actions can have on future projected biodiversity. It offers a more positive vision for the future than picturing the consequences of further biodiversity losses only, and can be applied to PD. We thus expect that targeting net positive outcomes for PD would encourage wider engagement in PD conservation.”

R1.5. 1111. “...both biomes..” Do you mean terrestrial and aquatic?

Considering that conservation actions (policies, management, etc) are taken at national/regional levels, and the discussion for terrestrial and marine conservation agenda is separate, do you feel it is adequate to analyze them together?

Reply to R1.5. We meant terrestrial and aquatic (marine + freshwater). We changed the sentence to make this clearer, it now reads as follows (lines 156-157):

“However, none of these considered all biomes (i.e. terrestrial, freshwater and marine) in the same analysis”.

We thank you for raising this critical point (= should we consider all mammals together or separate marine and continental mammals?), which was central in our discussions in the preparation of this study. In fact, the importance of the scale of conservation is the main issue of recent studies on the prioritisation of phylogenetic diversity (e.g.^{1,2}), and they clearly show that a global approach is largely more appropriate to maximise gains and to detect major losses of biodiversity. This indeed calls for the importance for considering conservation globally, and for the need for searching for new ways of considering the responsibilities of nations for conservation. The same applies for the consideration of different biomes. Therefore, while we recognize that conservation actions and agendas are separate for the continental and the marine biomes, we believe it is conceptually wrong to claim defining priorities for conserving **global** mammalian PD while focusing only on terrestrial (or marine, or freshwater) mammals. If priorities are defined for conserving global mammalian PD, then all mammals from all biomes must be considered together. By doing so, we identified priority species in the different biomes, showing that species from all biomes are important to protect global phylogenetic diversity of mammals – which would have been overlooked if biomes had been treated separately. We hope our study will contribute to thinking more globally about conservation responsibilities in the post-2020 framework. We now have integrated this issue in the discussion, lines 541-558:

“Here, we defined conservation priorities with one clear objective: maximising gain in expected PD for mammals globally. For this reason, we considered mammals from the three different types of biomes (i.e. terrestrial, freshwater and marine) together – despite the fact that conservation actions and agendas are separate for the continental (i.e. terrestrial and freshwater) and the marine biomes. Consequently, as there are less mammal species in the sea than on land, the marine environment is therefore not represented in our priority areas. Nevertheless, 22% of marine mammals are priority species – including all Sirenias – and priority species represent a high proportion of total species richness in the northern parts of the Atlantic and Pacific Oceans and in the Caspian Sea.”

¹Pollock, L.J., Rosauer, D.F., Thornhill, A.H., Kujala, H., Crisp, M.D., Miller, J.T., McCarthy, M.A. (2015). Phylogenetic diversity meets conservation policy: small areas are key to preserving eucalypt lineages. *Philosophical Transactions of the Royal Society B: Biological Sciences*, 370(1662), 20140007–20140007.

²Pollock, L.J., Thuiller, W., Jetz, W. (2017). Large conservation gains possible for global biodiversity facets. *Nature*, 546(7656), 141–144.

R1.6. I missed some more discussion among the results. For example, it seems, by your results, that Australia does not stand out in any of the results. It is surprising considering the evolutionary distinctiveness and threat of its mammalian diversity. Why is that?

Reply to R1.6. Indeed, Australia displays an important evolutionary distinctiveness for mammals. But the priority areas defined here exhibit, in addition of high evolutionary distinctiveness, relatively high levels of threatened species richness (see Figure S4 : correlation between GexpPD and TSR). Although some Australian mammals are highly threatened, Australia has a relatively low number of threatened species compared to other areas, both in total number of species and in proportion of its total species richness (see Figures S6b and S7b). This explains why no priority area is found in Australia.

The loss-significant areas defined here exhibit, in addition of high evolutionary distinctiveness, relatively low levels of threatened species richness but high species richness (see Figure S5: correlation between LexpPD and TSR, and LexpPD and SR). Although Australia exhibits low levels of threatened species richness (see Figure S7b), the fact that species richness in Australia is low (see Figure S6a) explains why no loss-significant area is found in Australia. However, Australia concentrates a high proportion of TOP LEDGE species (see Figure S3b), i.e. typically currently secure, distinctive species whose loss would mean a big loss in expected PD, regardless of their conservation status. We now discuss this in the revised version as follows (lines 315-321):

“The 1369 loss-significant species identified here are concentrated in Central America, Amazonia, Central Africa, and South-East Asia (Fig. 3b). They represent a high proportion of total species richness in the northern parts of Atlantic and Pacific Oceans, in the Black Sea, in South America, North Africa, Madagascar and Australia (Fig. S3b). This is because a lot of mammal species in these areas are loss-significant species, such as the humpback whale *Megaptera novaeangliae* in the Black Sea (TOP 1119 LEDGE) and the platypus *Ornithorhynchus anatinus* (TOP 14 LEDGE) in Australia.”

R1.7. 1273 "This low overlap between priority areas identified here and those previously-identified means that the priority areas identified here have few chances to be detected by other approaches"

That sentence made me very intrigued, because you did not actually overlap your priorities with the others and, as far as I remember, I am not sure if they are largely different from yours. Also, What precisely your framework is getting that others are missing? Make that clear.

Reply to R1.7. We did overlap our priorities with previously identified ones based on other metrics (lines 397-432, overlap results are shown in Table S3) and made a visual comparison with those identified by spatial conservation planning (lines 432-437). As explained in a reply to a previous comment (R1.3), the priorities we defined here focus on maximising **EXPECTED (ie future projected)** PD while other, previous ones focus on maximising **CURRENT EXTANT** biodiversity (whether on species or phylogeny-based metrics).

Reviewer #2 (Remarks to the Author):

1-What are the major claims of the paper?

Robuchon et al. aimed to inform global mammalian conservation based on the gain in expected PD if

priority species are protected. This was done using HEDGE and LEDGE scores. They showed that priority species and areas for conservation not only do not match the already-identified species and areas based on EDGE scores but are not well captured within the existing network of protected areas. The rationale for the study can be summarized in 4 points:

- There is no information on HEDGE and LEDGE scores for global mammalian diversity (L107-108).
- Although several studies have identified PD-based priority areas for mammals, none of them considered simultaneously both terrestrial and aquatic mammals,
- Of these studies, only one regional study has applied the expected PD framework of the HEDGE and LEDGE scores to identify priority sites (L109-113).
- Finally, a strategy requiring net outcome for PD would encourage wider engagement in PD-driven nature conservation (L114-119).

2-Are they novel and will they be of interest to others in the community and the wider field?

Setting priority for conservation is not novel; however, the approach used in the study is different from most studies: As opposed to most prioritisation strategies that seek to avoid further biodiversity losses, Robuchon et al. focused on expected gain of PD if priority species and areas are protected. In Line 114-119, they claim that their approach would encourage wider engagement in PD-driven nature conservation. The problem with this claim is that since 1992 that PD has been proposed, it hasn't been adopted widely in on-the-ground conservation actions and even in biodiversity policies as the authors themselves rightfully reported in L76-83.

R2.1. Does this suggest that decision makers do not believe in the added value of PD in conservation? One may think so (I do not share that view). So, how can one be sure that HEDGE approach here would avert this, i.e. convincing decision makers to act otherwise? Some even argue that PD seems to be more academic exercises than anything else. I suggest that the authors reword their claim that the use of HEDGE and LEDGE would encourage wider engagement in PD conservation.

Reply to R2.1.

You are right in that most of conservation actions are still very conservative, and that species richness, threat, vulnerability are still preferable currencies than Phylogenetic Diversity, as well as the consideration of evolution as reality in many countries. In spite of this, global agendas like IPBES, integrated the phylogenetic diversity in their Nature Contributions to People. A group to which some of the co-authors are taking part, the IUCN Species Survival Commission Phylogenetic Diversity Task Force is working now to implement targets and indicators of phylogenetic diversity in the post-2020 framework. So, our contribution will certainly be a reference for the integration of phylogenetic diversity in the conservation agendas.

That said, we think there has been some misunderstanding here. We believe our approach can encourage wider engagement in PD conservation not because it is based on PD (the value of PD is already appreciated, although this does not translate yet in international biodiversity policies) but because it is a way to reach net positive outcomes (gains>losses) for PD, by maximising gains and minimising losses in expected PD. We actually borrowed the argument from Bull et al. 2019, who wrote about biodiversity in general that “A strategy requiring net positive outcomes — above and beyond targets for preventing further declines— would encourage wider engagement in nature conservation”, and applied it to PD. We do believe that such positive vision (= if you follow these

conservation recommendations, it will result in a net gain in expected biodiversity – here PD) is more engaging than visions that only focus on avoiding biodiversity losses. We reworded the corresponding text in the introduction to make it clearer. It now reads as follows (lines 181-207):

“Despite the increasing awareness that PD conservation would benefit society by maintaining its options and the recognition that it should be explicitly included into global conservation goals²⁸, the preservation of PD is not yet implemented in international biodiversity policies - such as the strategy to 2020 of the UN Convention on Biological Diversity (CBD)²⁹ and the new EU biodiversity strategy for 2030³⁰. These strategies include evolution explicitly only via the preservation of genetic diversity for cultivated plants, and farmed and domesticated animals. A few months away from the 15th Conference Of the Parties of the CBD which will set the new conservation targets to 2030, the first drafts of the post-2020 biodiversity strategy³¹ overlook genetic diversity³² and do not yet incorporate PD. A recent proposal to track two PD indicators in the CBD post-2020 framework¹⁶ includes the IPBES existing indicator for expected PD loss, but does not suggest any indicator that would highlight expected PD gains. To fill this gap and contribute to promote PD conservation within the post-2020 framework, we propose to adapt Bull et al.³³'s suggestion to move beyond strategies seeking to avoid further biodiversity losses and develop strategies resulting in net positive outcomes for biodiversity to the particular case of PD. Indeed, strategies targeting net positive outcomes for biodiversity – i.e. biodiversity gains superior to biodiversity losses – would encourage wider engagement in biodiversity conservation³³, by highlighting the positive effects conservation actions can have on future projected biodiversity. It offers a more positive vision for the future than picturing the consequences of further biodiversity losses only, and can be applied to PD. We thus expect that targeting net positive outcomes for PD would encourage wider engagement in PD conservation.”

3-If the conclusions are not original, it would be helpful if you could provide relevant references.

As expected, they reported a high correlation between HEDGE and EDGE scores ($\rho = 0.94$) leading to the finding that 87% of HEDGE priority species are also EDGE priority species (L244-245).

R2.2. Interestingly, priority areas based on HEDGE overlap poorly with traditional biodiversity hotspots (L260-269). The authors claim that this low overlap is the good reason why HEDGE based conservation should be embraced in international biodiversity policies (L275-279). This argument needs to be strengthened. Conservation resources are already limited, traditional conservation measures based on SR are not always well implemented due to pressing development priorities in most species-rich countries, the PD since 1992 is not yet embraced in on-the-ground conservations nor in biodiversity policy, etc. That's why the argument consisting of saying the newly delimited priority species or priority species need to be embraced since they are different from the traditional ones doesn't seem to be convincing or exciting to me, although I fully agree that we need to be calling for PD to be included in biodiversity policy and actions. I am expecting more practical conservation implications than this, e.g. do we need to expand network of PA or Do we need to re-design the network to include the new priority areas and species? etc.

Reply to R2.2. We removed the argument consisting of saying that the newly delimited priority species or priority species need to be embraced since they are different from the traditional ones. Instead, we insist on the fact that “While it has been previously shown that species data can be good surrogates of PD for spatial conservation planning when the aim is to maximise current extant PD²³, our work suggests this is not true when the aim is to maximise gains in expected PD” (lines 440-443). The expansion of the PA network is discussed in the dedicated paragraph just after this one.

4-Is the work convincing, and if not, what further evidence would be required to strengthen the conclusions?

R2.3. The work is convincing as findings are well supported by the analysis of data collected, except that I do not get the rationale of the authors when they claim that the use of HEDGE will "widen the engagement of decision-makers in conservation".

Reply to R2.3. This has now been made clearer in the introduction (see reply to comment R2.1).

5-On a more subjective note, do you feel that the paper will influence thinking in the field?

R2.4. I have no doubt that the paper will draw attention of conservation biologists especially in the academic environment.

Reply to R2.4. Great, we hope so too!

6-Please feel free to raise any further questions and concerns about the paper.

R2.5. L129 In contrast to previous works (plural for work)

Reply to R2.5. We corrected the text accordingly.

R2.6. L162 Fig 3a shows west, central, east and southern Africa and not just South Africa.

Reply to R2.6. We corrected the text accordingly.

R2.7. L241mammalian PD for comparatively little effort.

Reply to R2.7. We corrected the text accordingly.

R2.8. L250of the TOP 25% HEDGE species, some... (comma after species)

Reply to R2.8. We corrected the text accordingly.

R2.9. L259-260 What is the implication for conservation?

Reply to R2.9. We reworded the text to better insist on conservation implications as follows (lines 420-425):

“Specifically, only a few priority areas are located in Amazonia (they are rather concentrated in South-East Africa, Madagascar, and South-East and Central Asia), while Amazonia concentrates hotspots of SR, threatened SR, PD and threatened PD (i.e. PD in the set of threatened species). In other words, while protecting Amazonia is a good strategy to maintain high levels of SR and PD, maximising gains in expected PD would better be achieved by protecting South-East Africa, Madagascar, and South-East and Central Asia.”

R2.10. L293. not clear to me. In the sentence above (L291-92), you said only 18.3% of priority areas is covered by the network but here you say the protection coverage is 0-99.9%. confusing.

Reply to R2.10. 18.3 % correspond to the percentage of the **TOTAL** surface of priority areas under protection (i.e. **over all priority areas**) while 0-99.9% correspond to the percentage of the surface under protection **within each priority area**. This has been clarified as follows (lines 508-511):

“The need of urgent conservation attention for some priority species also holds true for most priority areas as, overall, only 18.3% of the total surface of the priority areas (i.e. over all priority areas) is covered by the existing global network of protected areas. The protection coverage of the priority areas (i.e. within each priority area) ranges from 0% to 99.9%.

R2.11. L302-303: I agree fully.

Reply to R2.11. Happy to read this!

R2.12. L305: Figure 6. This spatial overlap should be done and analyzed for both HEDGE and LEDGE. It's important to see the extent of coverage of LEDGE as well in the existing network of PAs.

Reply to R2.12. We added a figure 6b showing the protection coverage of loss-significant areas and wrote a few lines about the results (lines 520-526) :

“Interestingly, loss-significant areas are better protected than priority areas, as 32.5% of the total surface of the loss-significant areas (i.e. over all loss-significant areas) is covered by the existing global network of protected areas. Nonetheless, we stress out that some loss-significant areas are poorly protected, notably in some parts of Amazonia and Central Africa (Fig. 6b). Loss-significant areas in these regions are good candidates to add new, non-necessarily strictly protected areas or to take other area-based conservation measures to prevent losses in expected PD.”

To be consistent, we also moved the text regarding protection of loss-significant species earlier in this section: (lines 480-483):

“The majority of the loss-significant species do not benefit from any conservation measure (54%), including the aardvark *Orycteropus afer* (TOP 1 LEDGE); and for the species benefitting from at least one conservation measure, land/water management is the most common measure (Fig. 5b, Table S5)”.

We also changed the title of the corresponding section in “Current protection and guidance for future management of the newly-identified species and areas of interest” so it better reflects the updated content.

R2.13. L310-327 I do not see the EU recommendation to eradicate this *M. coypus* (or any other alien invasive species) as a conservation paradox, in light of your recommendations to avoid human actions that might damage this species. I think your recommendation to avoid human actions on priority species should be more focused on priority species in their native range i.e. new human actions that may damage species in their native range should be avoided. Alien invasive species are themselves threatening biodiversity in their recipient environment. By avoiding any intervention to eradicate or control alien invasive species in their recipient environment, the risk of losing local diversity is high. I suggest that conservation measures of problematic species like alien should focus on native ranges. Or when in non-native range, they should be in controlled environment, e.g. Zoos.

Reply to R2.13. We reworded this part of the text (which has now been integrated in the section “Current protection and guidance for future management of the newly-identified species and areas of interest”) to remove the words “conservation paradox” and highlight that management of species that are both of global conservation interest and locally or regionally invasive should vary across the species range (lines 483-501):

“Interestingly, 2% of the loss-significant species – of which 10 species are also priority species – have populations which have been introduced outside of their native range (Fig. 5b, Table S5). This includes the platypus *Ornithorhynchus anatinus* (TOP 14 LEDGE), native from Eastern Australia and which has been introduced further west in Kangaroo Island (Australia). This also includes the coypu *Myocastor coypus* (TOP 98 LEDGE), native from South America, which has been introduced to North America, Europe, Africa, and Asia and is considered invasive in some countries of introduction (*M. coypus* is on the list of invasive alien species of Union concern). While our work recommends that each new human action that might damage this species should be avoided, the regulation of the European Union on invasive alien species imposes to take measures for its rapid eradication. This example highlights that some species can be, at the same time, of global conservation interest and locally or regionally concerning because of their invasiveness, highlighting the need of global evaluation in order to seize this kind problem at different part of their ranges. In such situation, species management should vary across the species’ range so that native populations remain (or become) healthy while introduced populations are controlled (or even eradicated). Although we did not make a comprehensive assessment of which species of global conservation interest identified here have become invasive, we believe this is an issue worth of further investigation.”

R2.14. L347-350 I agree fully with this type of argument.

Reply to R2.14. Happy to read this!

R2.15. L364: EDGE (not Edge?)

Reply to R2.15. We corrected this.

R2.16. 370-371 "...has the power to widen the engagement of decision-makers in PD conservation". I do not buy this argument as I indicated above. The integration of evolution in conservation is lacking visibly buy-in from decision- or policy-makers. I don't see how the use of HEDGE, as promoted in this study, will change this.

Reply to R2.16. As explained in reply to R2.1, the use of the HEDGE and LEDGE approaches for species and areas is a way to reach net positive outcomes in PD. This net positive outcome strategy, whatever the biodiversity metric it applies to, is a way to widen the engagement of decision-makers in biodiversity conservation – because it offers a more positive vision for the future than the prevention of further biodiversity losses only. We reworded this part of the conclusion to make our point clearer, and it now reads as follows (lines 557-564):

“The strategy we used here to identify priorities for conserving global mammalian PD directly reflects the opportunity to increase expected PD. Implementing it in combination with preventive conservation targeting loss-significant species and areas in multiple taxonomic groups, i.e. maximising gains and minimising losses of expected PD, is a proactive way to reach a conservation target of net positive outcomes for PD. Because it reflects the positive effects conservation actions can have on expected PD, such a conservation target offers a more positive vision for the future than picturing the

consequences of further PD losses only. This may therefore help to widen the engagement of decision-makers in PD conservation.”

7- We would also be grateful if you could comment on the appropriateness and validity of any statistical analysis, as well the ability of a researcher to reproduce the work, given the level of detail provided.

R2.17. The data analysis including the statistics is great, and I can't see any flaw on this regards. The R script used is not publicly available yet but will be after the paper is published.

Reply to R2.17. Thanks, happy to read this! Please also note that all the code to make the analyses from scratch and generate the figures is currently stored in a GitHub repository which will be made publicly available upon acceptance of the manuscript (or sooner if one reviewer requires it).

Reviewer #3 (Remarks to the Author):

Referee's report for Manuscript 266789 (Robuchon et al.)

The core of this study is a consideration of which species of mammal (and in aggregate, which places where they occur) whose conservation (their prob(extinction) is set to 0) would safeguard most "expected" (aka projected future) phylogenetic diversity, or whose loss (their prob(extinction) set to 1) would lead to the greatest loss of expected phylogenetic diversity. The ingredients for this exercise are a dated phylogenetic tree of all mammals (from Phylacine 2.0), a Red List index from the IUCN converted to a p(ext), and a map of the species extents of occurrence.

There is a great deal to commend in this work, and it represents a large investment in time and thought.

R3.1. I think the conceptual switch presented in figure 1, where one considers species from the contrasting points of view of how they contribute to future diversity (the HEDGE metric), or how their loss might contribute to future loss of diversity (the LEDGE metric) is elegant.

Reply to R3.1. Thanks, this is much appreciated!

That said, I have two concerns with the paper at a high organizational and data-stream level, neither likely to be welcome to the authors.

R3.2. The set-up of this paper is decidedly political – the authors are explicit that this exercise is in aid of convincing policymakers to consider PD in the policy being created for the revision of the Convention on Biological Diversity in 2021. While this is an admirable (if perhaps still premature) goal (at least till some new papers see the light of day), it does mean one has to particularly conservative.

Reply to R3.2. We have a point of view different from yours for two reasons. First, a recent paper that we cited in this revised version (Diaz et al., 2020, *Set ambitious goals for biodiversity and sustainability*, Science) made a strong case that PD must be integrated into the CBD post-2020 framework – we therefore think that the timing for our paper is particularly adequate. Second, we do not think that science needs to be conservative to be sued in policy. This is especially true for conservation, which has been early described as a “crisis discipline” by Soulé in his paper entitled

“What Is Conservation Biology” from 1985: “Conservation biology [...] is a crisis discipline. A conservation biologist may have to make decisions or recommendations about design or management before he or she is completely comfortable with the theoretical and empirical bases of the analyses”. This is why, while we reckon our recommendations may need to accommodate new scientific knowledge in the future years (including updated phylogenies and alternative model to calculate probabilities of extinction), we believe our work is mature enough to be integrated into policy-making.

R3.3. I start with the easier or the two issues. The Phylacine 2.0 tree produces fairly different DR scores than do point estimates from the newer and better-dated Upham et al. (2019) trees of mammals (see figure 6(b), middle panel from that paper, which compares these numbers directly). $DR=1/ES$, and ES and ED are generally strongly positively correlated (in my experience, with $Rho \gg 0.7$); both are strongly dependent on the pendant edge length, and that is the edge that also contributes most (by quite a ways) to HEDGE and LEDGE calculations. I think we need to know how sensitive the results in this paper are to the particular tree used: correlations across common species for the two (sets of) trees would be welcome, and at the very least, we need to know if the top 100 species are the same.

Reply to R3.3. To test how sensitive our results are to the particular tree used, we re-calculated HEDGE and LEDGE scores with the phylogeny of Upham et al. (2019). We then carried out correlations tests between species scores calculated with the Phylacine phylogeny and species scores calculated with the Upham phylogeny (see Figure S1 of the Supplementary Material), and we also compared TOP 25% species between the two phylogenies. We found that HEDGE scores are highly robust and LEDGE scores moderately robust to recent phylogenetic updates. We updated the text in the different sections of the manuscript:

- lines 259-263: “These HEDGE scores calculated with the phylogeny from PHYLACINE³⁴ are highly correlated to those calculated with the phylogeny of Upham et al.³⁵ ($\rho = 0.86$, Fig. S1). Moreover, 93% of the priority species identified with the phylogeny from Upham et al.³⁵ are also identified as priority species using the phylogeny from PHYLACINE³⁴. This suggests that HEDGE scores are highly robust to recent updates of the mammalian phylogeny.

- lines 304-314: “These LEDGE scores calculated with the phylogeny from PHYLACINE³⁴ are moderately correlated to those calculated with the phylogeny of Upham et al.³⁵ ($\rho = 0.54$, Fig. S1). Fifty five percent of the loss-significant species identified with the phylogeny from Upham et al.³⁵ are also identified as loss-significant species using the phylogeny from PHYLACINE³⁴. This suggests that, contrary to HEDGE scores, LEDGE scores are only moderately robust to recent updates of the mammalian phylogeny. A possible explanation is that HEDGE scores seem to be more driven by extinction risk than LEDGE scores (Fig. S2), and therefore less impacted by phylogenetic changes. With evolving knowledge, results will have to be regularly updated and recommendations may evolve accordingly. In this regard the robustness of HEDGE to phylogeny updates is encouraging but the more moderate robustness of LEDGE and its consequences for policy making still need to be investigated.”

- lines 712-717: “To test if our results were robust to recent updates of the mammalian phylogeny, we replicated our calculations of HEDGE and LEDGE scores using the phylogeny of Upham et al.³⁵. We then tested the correlation between scores calculated with the phylogeny from PHYLACINE³⁴ and those calculated with the phylogeny of Upham et al.³⁵ using a Spearman correlation test. We also compared the priority species and the loss-significant species identified with the two phylogenies.

R3.4. The second issue is thornier. A recent paper based on modelling work by Folmer Bokma (Monroe et al., *Biol. Lett.* 2019) estimated the instantaneous $p(\text{ext})$ of all birds over the past 30 years using Red List status changes, probabilities that can be projected over any time period (eg. 50 years). These data-based estimates are quite different from those derived from Red List definitions. For instance, their $p(\text{ext})$ over 10 years for a species designated as CR seems to be less than a tenth of the Red List definition: 0.04 at the upper limit (or 4%, $= 1 - (1 - 0.0039)^{10}$, taking the values from their supplementary table 2, which is supposed to ignore active downlisting, vs. the value of 0.5 pegged by the IUCN and used here). The corresponding Monroe et al. value for EN is less than 1/3 the Red List definition, so the differences are not due to a simple multiplier. This is because the Red List definitions are based on "no intervention" while the actual transition rates are estimated from observed changes that incorporate all that is done and not done for species on the list. Conservation attention changes a species' outlook, and this outlook changes with status, both which make sense. However, this suggests that forward projections based on the definitions are just plain wrong, because the projections are based on the assumption that we have, as of the time of writing, stopped all conservation activity. The authors of the Monroe et al. paper do not comment on this anywhere and seem unaware. Regardless, to the extent my comparison here is correct, projections created from Red List definitions (e.g. from Mooers et al. 2008 or the updated ones from Davis et al. PNAS 2018) are suspect.

Of course, the main results here (the species rankings) might be robust to the distribution of $p(\text{ext})$ used. Documenting that would increase the strength of the argument that HEDGE and LEDGE are powerful metrics of diversity protection. But until we know this, I do not advocate using expected PD in a policy-facing framework with Red-List-derived $p(\text{ext})$, as the conclusions might be misleading.

Reply to R3.4. We agree regarding the fact that using IUCN-based $p(\text{ext})$ implies that projections of future PD are calculated considering that "we stopped all conservation at the time of writing" – or to be even more accurate, that "the risk categories of the species at the time of writing this manuscript do not change" (because even if there are some conservation actions, they may not succeed in improving the conservation status of a species if it's already in a vortex of extinction). Actually, we now wrote this clearly in the legend of Figure 1: the grey circle representing expected PD stands for "no change in species extinction risk". But every projection is based on assumptions, so, as long as we specify our assumptions, we do not see why this is a problem. In addition, we are not interested by the projections of expected PD under "no change in species extinction risk" per se, but by the difference between expected PD under "no change in species extinction risk" and expected PD under complete protection ($p_{\text{ext}} \text{ go to } 0$: HEDGE for species, G_{expPD} for areas)/expected PD under complete extinction ($p_{\text{ext}} \text{ go to } 1$: LEDGE for species, L_{expPD} for areas). Finally, what is interesting for policy making here is not just the priorities and loss-significant species/areas identified, but how we identify them using a net positive outcome approach. Such net positive outcome approach can be applied with all possible methods to calculate $p(\text{ext})$.

Although we reckon that alternative frameworks could be used to calculate $p(\text{ext})$, we therefore think there is no problem in using Red-List-derived $p(\text{ext})$ in our paper.

We have added a few sentences about this issue in the "limits and perspectives" section (lines 543-552):

"Finally, we used IUCN designations projected to 50 years to derive probabilities of extinction from Red List extinction risk categories⁴⁹ for each species and calculate expected PD in 2070. It means that

our baseline scenario for 2070 corresponds to the remaining PD we would observe if the risk categories of the species at the time of writing this manuscript do not change. The species and areas of interest we defined correspond to those whose protection would maximise gains (priority species/areas) or minimise losses (loss-significant species/areas) in expected PD compared to this baseline scenario (Fig. 1). As new models are being developed to estimate probabilities of extinction, such as a recent one considering the past dynamics of species through IUCN Red List extinction risk categories⁵⁰, we note that our approach could be extended to any baseline scenario using such new models.”

In the interests of time, I will stop my report here, in the hopes of getting a second version – if not at this journal, somewhere else.

R3.5. I have made a number of suggestions to the first part of the paper in MS Word and have included that as an attachment to this review.

Reply to R3.5. We took all these suggestions into account in this revised version, except:

- The reference to the “tree of life” in the abstract (we would like to limit the vocabulary either to PD or evolutionary history)
- The edit at the beginning of the abstract (because we restructured and shortened the abstract considerably)

R3.6. Overall, I think the argument that a HEDGE perspective is somehow a more useful "net outcome" might be compelling, but needs to be presented more clearly and in one place – currently it is spread across several different paragraphs.

Reply to R3.6. We have clarified this net positive outcome approach and why it is more engaging for decision-makers in one place in the introduction (lines 181-207):

“Despite the increasing awareness that PD conservation would benefit society by maintaining its options and the recognition that it should be explicitly included into global conservation goals²⁸, the preservation of PD is not yet implemented in international biodiversity policies - such as the strategy to 2020 of the UN Convention on Biological Diversity (CBD)²⁹ and the new EU biodiversity strategy for 2030³⁰. These strategies include evolution explicitly only via the preservation of genetic diversity for cultivated plants, and farmed and domesticated animals. A few months away from the 15th Conference Of the Parties of the CBD which will set the new conservation targets to 2030, the first drafts of the post-2020 biodiversity strategy³¹ overlook genetic diversity³² and do not yet incorporate PD. A recent proposal to track two PD indicators in the CBD post-2020 framework¹⁶ includes the IPBES existing indicator for expected PD loss, but does not suggest any indicator that would highlight expected PD gains. To fill this gap and contribute to promote PD conservation within the post-2020 framework, we propose to adapt Bull et al.³³'s suggestion to move beyond strategies seeking to avoid further biodiversity losses and develop strategies resulting in net positive outcomes for biodiversity to the particular case of PD. Indeed, strategies targeting net positive outcomes for biodiversity – i.e. biodiversity gains superior to biodiversity losses – would encourage wider engagement in biodiversity conservation³³, by highlighting the positive effects conservation actions can have on future projected biodiversity. It offers a more positive vision for the future than picturing the consequences of further biodiversity losses only, and can be applied to PD. We thus expect that targeting net positive outcomes for PD would encourage wider engagement in PD conservation.”

We also repeat these arguments in our conclusion section, both because we think it is important to remind the reader why this approach may be more appealing for decision-makers in this concluding section, and because reviewer 2 asked us to clarify this part of the conclusion (lines 557-568):

“The strategy we used here to identify priorities for conserving global mammalian PD directly reflects the opportunity to increase expected PD. Implementing it in combination with preventive conservation targeting loss-significant species and areas in multiple taxonomic groups, i.e. maximising gains and minimising losses of expected PD, is a proactive way to reach a conservation target of net positive outcomes for PD. Because it reflects the positive effects conservation actions can have on expected PD, such a conservation target offers a more positive vision for the future than picturing the consequences of further PD losses only. This may therefore help to widen the engagement of decision-makers in PD conservation. We thus hope that our work will, together with other initiatives such as the IUCN Species Survival Commission Phylogenetic Diversity Task Force, inform the CBD’s Scientific Subsidiary Body and encourage the inclusion of a conservation target requiring net positive outcomes for PD in the CBD post-2020 framework.”

R3.7. Further, I think the all-important discussion of expected PD, EDGE, HEDGE and LEDGE could use some work. "expected PD" needs to be introduced first (perhaps as "projected future PD"). So, for example, as an unrequested potential edit:

Global PD-based priority mammal species have been identified^{8,9} using the EDGE (‘evolutionarily distinct and globally endangered’) approach to attribute a priority score to a species by combining its score of evolutionary distinctiveness (ED) with its status on the IUCN Red List of Threatened SpeciesTM as an estimate of its extinction probability. The EDGE score is a crude projection of how much evolutionary history is "expected" to be lost, species by species, if no action is taken, based on the assigned probability of extinction of that species. This framework can be extended to the entire tree: if every tip is given an extinction probability (e.g. over the next 50 years), one can project how much of the tree will persist into the future, or the "expected PD" of the tree^{12,13}. One can in turn assign values to individual species, either the contribution to this expected PD of a species if it is secured (its probability of extinction is set to 0), or the loss of expected PD if it were to go extinct (its probability of extinction is set to 1)¹¹. We refer to the first projection as a HEDGE¹¹ score, and re-christen the latter as a LEDGE¹⁵ score. We note that the globally important EDGE of Existence Programme^{7,8} is endorsing expected PD calculations¹³ as an alternative to its conventional EDGE scores¹⁴.

HEDGE scores have tentatively been assigned to aquatic mammals¹⁰ but here we generate HEDGE and LEDGE values across all terrestrial and marine mammals globally, compare them, identify species with the highest values, and map species globally to identify hotspots. We find XXXX...

Reply to R3.7. We have extensively reworded the text in the introduction to better present EDGE vs expected PD approaches (lines 148-173), and specifically have introduced “expected PD” in the introduction as follows (lines 152-157):

“While EDGE used a simple partitioning of the total PD of a clade among its member species, species of interest for conserving PD may also be identified based on how our actions on such species - assuming they produce some changes in their probabilities of extinction - influence future projected

PD. This framework, which uses probabilities of extinction to calculate future projected PD (hereafter expected PD), can cover any scenario of changes in probabilities of extinction¹³.”

REVIEWER COMMENTS

Reviewer #1 (Remarks to the Author):

The authors addressed all my concerns properly. I am satisfied with the modifications that they made.

Reviewer #2 (Remarks to the Author):

This is a well written report that can inform the improvement of the post2020 CBD framework, specifically with regards to genetic diversity. I am happy with how my initial points/concerns/comments have been addressed. I do not have further comments or concerns to raise. I am looking forward to the R script to be made public.

Prof. Kowiyou Yessoufou
University of Johannesburg
South Africa

Reviewer #3 (Remarks to the Author):

To contextualise my comments, I first organize these various acronym-heavy metrics for the editor. The authors all know this literature well, so I have not included a citation list.

More than 25 years ago, Faith introduced a species-specific diversity metric, defined as the phylogenetic contribution a species makes to the total phylogenetic diversity of a set of species (Faith, 1992). In 2007, Isaac et al. of the ZSL published the species-specific "EDGE" metric, an ad-hoc combination of a diversity metric they coin evolutionary distinctness (ED) and risk of extinction, and this group has since published several papers on ranking mammal species and global hotspots based on this metric (Isaac et al., PloS One 2007; Collen et al., Phil Trans 2011; Safi et al., PloS One 2013). EDGE is just a log-transformed "expected loss," characterised by Redding and Mooers (Biol. Cons.2006) as the product of a species' "evolutionary worth" and its probability of extinction ($p(\text{ext})$). This concept of a metric based on expected loss was formalized by Steel et al. in 2007 (following Haake et al., 2008): Steel et al proposed two metric: the expected gain in future evolutionary history if a species is secured, and the expected loss of future evolutionary history if a species goes extinct; importantly, they referred to both as "HEDGE" metrics, their sum being a species' contribution to future phylogenetic diversity under the status quo. Also in 2007, and in response to the ad hoc diversity metrics introduced by Isaac et al. and Redding and Mooers, Faith drew on the "expected PD" framework of Witting and Loeschke 1995 to propose "expected PD gain" – again, the expected gain in future evolutionary history if a species is secured -- as a metric for ranking species for conservation (Faith 2007).

In 2012, Faith separated out the two HEDGE metrics, referring to the expected gain version as HEDGE,

and christening the latter, expected loss, version, as LEDGE. This naming convention seems intuitive. The two metrics highlight subtly different measures of conservation worth: species currently not likely to contribute to future evolutionary history because they are not expected to survive, but whose conservation would secure a large portion of the future evolutionary tree (HEDGE), and species that are not expected to be lost, but, if they were to be, would take a lot of evolutionary history with them (LEDGE). Though both indices use the same future phylogenetic tree, because HEDGE is a function of $p(\text{ext})$ and LEDGE is a function of $1-p(\text{ext})$, the two measures need not rank the same species highly.

The expected PD gain version of the HEDGE metric is said to be the successor to EDGE in the ZSL Edge of Existence program. The present manuscript gets a jump on the ZSL here by publishing HEDGE and LEDGE scores for mammals, and identifying global hotspots based on these two metrics. I believe these are the first calculation of HEDGE and LEDGE for a major taxon. Given this, I am a bit disappointed that the ZSL is not formally involved in this study, but that sort of political consideration is beyond the scope of this report.

I start by raising two other "political" issues, given the framing the authors continue to pursue.

The language in the very recent (Oct. 2020) Diaz et al. 2020 précis from the IPBES, explicitly names "evolutionarily distinct species" and explicitly refers to "safeguard[ing] the Tree of Life." This is very encouraging as support for this area; the concept of prioritizing evolutionarily distinct species because they contribute more to evolutionary history seems to be gaining traction, with expected downstream policy consequences. This important reference supports the present study, and I thank the authors for making this clear.

Nonetheless, the language at line 198, that "to fill this gap and to contribute to promote PD conservation in the post 2020 framework, we [present HEDGE and LEDGE scores for all mammals]..." (I believe the authors mean either "contribute to promoting..." or "to continue to promote...") still seems a stretch for this paper (as does the statement later on: "We thus expect that targeting net positive outcomes for PD would encourage wider engagement in PD conservation."). I also note that EDGE is also a clearly-defined "expected loss" metric, and so this "net positive outcome" framing is more rhetorical than substantive.

To clarify: I do not believe these lists will fill any gap; in their current incarnation, they may simply widen it. I take up this issue below, in the context of returning to the two substantive questions I raised in my first short report.

I thank the authors for doing some critical re-analyses with the Upham et al. mammal tree to test the robustness of their rankings to new phylogenetic information. I am very encouraged to see that the top 50% of HEDGE scores of mammals are robust to these different phylogenies – as the authors state, $HEDGE_i$ is governed primarily by local $p(\text{ext})$ and local phylogenetic information (e.g. whether species i is monotypic or a member of a large genus). This is great news for the HEDGE metric.

However, the weak correlation for LEDGE across the two trees is problematic in the context of the framing of this paper (e.g., line 198 quoted above). Again, this instability is not that surprising upon reflection, since there is a much larger pool of (tree-specific) high ED species that are all low risk and so who vie to be high-ranking LEDGE species. But, either one of the two phylogenies is closer to the truth, or we don't know which species are LEDGE species. If it is the latter, then the explicit audience of this paper (policymakers) may continue to believe that energy spent on these indices is still energy misspent, and the creators and promulgators have more work to do. This is not to question the results, but rather the packaging (and with the packaging, perhaps the justification for a general audience journal in the Nature stable). I would be much less hesitant to recommend publication of this work if it were not so policy-facing.

To continue in this vein: in my first review, I highlighted that the nominal $p(\text{ext})$ (here, in a 50 year window) that is attached to a IUCN category is not in fact the $p(\text{ext})$ of a species assessed as belonging to that category, at least not if one considers actual change in species trajectories to extinction. The authors' reply, that this does not matter not too because one can just define these $p(\text{ext})$ to represent the situation if nothing were to be done from here on is technically correct, except that much is being done. This matters when policymakers are courted, actual species lists are presented, and particular species are highlighted. As an academic exercise, it would be very useful to see how robust HEDGE scores are to constant rank but changing $p(\text{ext})$ values; in a paper explicitly directed to the folks who might influence framers of international conventions, I believe it is critical. We simply do not know if the HEDGE lists and HEDGE areas are robust to the $p(\text{ext})$ used. (I do not believe we know this for EDGE lists either). I did not see any substantive response to this in the new version. I draw the authors attention to one reasonable way forward, most recently suggested by Gouhier and Pillai (Frontiers in Ecol and Evol, 2020), that phylogenetic information be used as an index to rank species within IUCN categories.

I can now turn to a few additional, new issues.

There are over 3000 species with reported HEDGE scores = 0.000, but these are given different ranks (ranging from 2200-5476.5); this oddity (or pathology) can be seen in the discontinuity in Figure S4 and the odd HEDGE-HEDGE plot in S1. These are likely zero-length pendant edge species in the Phylacine tree, but of course no mammal species is zero years old. This needs to be dealt with. The fact it was not suggests this paper was a bit rushed.

In expectation, EDGE and HEDGE scores can be quite different, but whether they are empirically would seem important for situating this paper in the literature. If we drop these low-rank, incorrect zero-branch species, the simple correlation coefficient between EDGE and $\log(\text{HEDGE})$ is ~ 0.80 (and same if one looks at the ranks only). But what if we look at only the top, say, 100 or 1000 species? And if high priority EDGE and high priority HEDGE species and areas of the world are not the same, which species or taxa or situations explain this? These questions take me back to the first political issue I raised above: given EDGE is the only active diversity-index conservation ranking system in place globally, and that it was introduced using mammals, it would seem natural that it be the baseline for considering HEDGE (and LEDGE) scores. As the authors know well, there are many diversity indices littering the literature.

We need good reasons to start dropping those that are redundant, unstable, hard to communicate, or that do not capture aspects of biodiversity that matter. (I note to the editor that this may be completely out of line.)

I note that the link in Ref. 16 (Owen et al. proposal to CBD) does not work. I do not know what this endeavour represents.

148: identified, not defined

205: I have never seen the term "picturing" used like this – please check.

220 (and elsewhere). This analysis of introduced species seems completely off-topic, especially if the goal is to engage with policy makers. It is a distraction and should be toned down or dropped.

Overall, some of the new text reads as if it was rushed; it could be tightened.

Finally, I apologize for the extreme tardiness of this review. I have no good excuse except pandemic-induced administrative hassles with my lab and my students, and lower productivity overall. I appreciate that the authors may disagree with some of my comments, but I hope they are helpful nonetheless.

Arne Mooers

POINT-BY-POINT RESPONSE TO REVIEWER COMMENTS

Important note: the line numbering corresponds to the revised version of the manuscript showing all track changes (as requested by the editor), but the text copied/pasted within this response to reviewers corresponds to the revised clean version (not showing changes). We hope these clarifications will make the revision process more easy-going.

Reviewer #1 (Remarks to the Author):

R1.1. The authors addressed all my concerns properly. I am satisfied with the modifications that they made.

Reply to R1.1. We are glad to read that! Thank you for your inputs which have helped us to improve the quality of our manuscript.

Reviewer #2 (Remarks to the Author):

R2.1. This is a well written report that can inform the improvement of the post2020 CBD framework, specifically with regards to genetic diversity. I am happy with how my initial points/concerns/comments have been addressed. I do not have further comments or concerns to raise. I am looking forward to the R script to be made public.

Prof. Kowiyou Yessoufou
University of Johannesburg
South Africa

Reply to R2.1. Thank you for your comments which contributed to make our paper better! The R script will be publicly available on GitHub – on this address:
<https://github.com/MarineRobuchon/consmampd> - as soon as the manuscript is accepted.

Reviewer #3 (Remarks to the Author):

To contextualise my comments, I first organize these various acronym-heavy metrics for the editor. The authors all know this literature well, so I have not included a citation list.

More than 25 years ago, Faith introduced a species-specific diversity metric, defined as the phylogenetic contribution a species makes to the total phylogenetic diversity of a set of species (Faith, 1992). In 2007, Isaac et al. of the ZSL published the species-specific "EDGE" metric, an ad-hoc combination of a diversity metric they coin evolutionary distinctness (ED) and risk of extinction, and this group has since published several papers on ranking mammal species and global hotspots based on this metric (Isaac et al., PloS One 2007; Collen et al.,

Phil Trans 2011; Safi et al., PloS One 2013). EDGE is just a log-transformed "expected loss," characterised by Redding and Mooers (Biol. Cons.2006) as the product of a species' "evolutionary worth" and its probability of extinction ($p(\text{ext})$). This concept of a metric based on expected loss was formalized by Steel et al. in 2007 (following Haake et al., 2008): Steel et al proposed two metric: the expected gain in future evolutionary history if a species is secured, and the expected loss of future evolutionary history if a species goes extinct; importantly, they referred to both as "HEDGE" metrics, their sum being a species' contribution to future phylogenetic diversity under the status quo. Also in 2007, and in response to the ad hoc diversity metrics introduced by Isaac et al. and Redding and Mooers, Faith drew on the "expected PD" framework of Witting and Loeschke 1995 to propose "expected PD gain" – again, the expected gain in future evolutionary history if a species is secured -- as a metric for ranking species for conservation (Faith 2007).

In 2012, Faith separated out the two HEDGE metrics, referring to the expected gain version as HEDGE, and christening the latter, expected loss, version, as LEDGE.

This naming convention seems intuitive. The two metrics highlight subtly different measures of conservation worth: species currently not likely to contribute to future evolutionary history because they are not expected to survive, but whose conservation would secure a large portion of the future evolutionary tree (HEDGE), and species that are not expected to be lost, but, if they were to be, would take a lot of evolutionary history with them (LEDGE). Though both indices use the same future phylogenetic tree, because HEDGE is a function of $p(\text{ext})$ and LEDGE is a function of $1-p(\text{ext})$, the two measures need not rank the same species highly.

The expected PD gain version of the HEDGE metric is said to be the successor to EDGE in the ZSL Edge of Existence program. The present manuscript gets a jump on the ZSL here by publishing HEDGE and LEDGE scores for mammals, and identifying global hotspots based on these two metrics. I believe these are the first calculation of HEDGE and LEDGE for a major taxon. Given this, I am a bit disappointed that the ZSL is not formally involved in this study, but that sort of political consideration is beyond the scope of this report.

I start by raising two other "political" issues, given the framing the authors continue to pursue.

The language in the very recent (Oct. 2020) Diaz et al. 2020 précis from the IPBES, explicitly names "evolutionarily distinct species" and explicitly refers to "safeguard[ing] the Tree of Life." This is very encouraging as support for this area; the concept of prioritizing evolutionarily distinct species because they contribute more to evolutionary history seems to be gaining traction, with expected downstream policy consequences. This important reference supports the present study, and I thank the authors for making this clear.

R3.1. Nonetheless, the language at line 198, that "to fill this gap and to contribute to promote PD conservation in the post 2020 framework, we [present HEDGE and LEDGE scores for all mammals]..." (I believe the authors mean either "contribute to promoting..." or "to continue to promote...") still seems a stretch for this paper (as does the statement later on: "We thus

expect that targeting net positive outcomes for PD would encourage wider engagement in PD conservation."). I also note that EDGE is also a clearly-defined "expected loss" metric, and so this "net positive outcome" framing is more rhetorical than substantive.

To clarify: I do not believe these lists will fill any gap; in their current incarnation, they may simply widen it. I take up this issue below, in the context of returning to the two substantive questions I raised in my first short report.

Reply to R3.1. Here there is a misunderstanding regarding what will “fill this gap and contribute to promoting” (we indeed meant “contribute to promoting” and corrected the manuscript accordingly). In agreement with reviewer 3, we do not believe that the HEDGE and LEDGE scores for all mammals we present here will fill any gap. However, we do believe that applying the “net positive outcome” approach to PD will help to incorporate PD in biodiversity policies such as the CBD post-2020 framework. We reformulated our paragraph as follows (lines 106-110) : “To further contribute to promoting PD conservation within the post-2020 framework, we propose to discuss, for the particular case of PD, Bull et al.³⁵'s suggestion to move beyond strategies seeking to avoid further biodiversity losses and develop strategies resulting in net positive outcomes for biodiversity”. Our work here aims to reach such net positive outcome for PD, consisting of a prioritisation strategy to increase gains in expected PD and preventive conservation actions that would limit losses in expected PD, that we applied to mammals.

According to Isaac et al. (2007) who introduced the EDGE metric, “EDGE scores are therefore equivalent to a log-transformation of the species-specific expected loss of evolutionary history”. We thus agree that EDGE is an expected loss metric, although it does not directly reflect expected loss of phylogenetic diversity (noting that the EDGE group has explored “EDGE2.0” modifications to create links to expected loss of phylogenetic diversity). As Bull et al. in his paper proposing the net positive outcome framing, we recognise that the novelty in this framing compared to conservation targets that focus on avoiding losses is more due to language than to substance (and indeed, an avoided loss of biodiversity at t0 translates into a gain of expected biodiversity at t1). However, it does not make the framing less powerful, as formulating targets in terms of net positive outcomes can have major implications for the way in which conservation actions are delivered - by highlighting the positive effects conservation actions can have on future projected biodiversity. We now have made this recognition clearer in the manuscript (lines 103-107): “Although the novelty is mainly semantic, using a net positive outcome approach can have major implications for the way in which conservation actions are delivered because it highlights the positive effects conservation actions can have on future projected biodiversity”.

R3.2. I thank the authors for doing some critical re-analyses with the Upham et al. mammal tree to test the robustness of their rankings to new phylogenetic information. I am very encouraged to see that the top 50% of HEDGE scores of mammals are robust to these different phylogenies – as the authors state, HEDGE_i is governed primarily by local p(ext)

and local phylogenetic information (e.g. whether species *i* is monotypic or a member of a large genus). This is great news for the HEDGE metric.

Reply to R3.2. Thank you, this is great news indeed for the HEDGE metric!

R3.3. However, the weak correlation for LEDGE across the two trees is problematic in the context of the framing of this paper (e.g., line 198 quoted above). Again, this instability is not that surprising upon reflection, since there is a much larger pool of (tree-specific) high ED species that are all low risk and so who vie to be high-ranking LEDGE species. But, either one of the two phylogenies is closer to the truth, or we don't know which species are LEDGE species. If it is the latter, then the explicit audience of this paper (policymakers) may continue to believe that energy spent on these indices is still energy misspent, and the creators and promulgators have more work to do. This is not to question the results, but rather the packaging (and with the packaging, perhaps the justification for a general audience journal in the Nature stable). I would be much less hesitant to recommend publication of this work if it were not so policy-facing.

Reply to R3.2. We recognise that, because of the weak correlation for LEDGE across the two trees, the identity of LEDGE species may further change with new phylogenetic knowledge. This is the case for many biodiversity indices that change with new scientific knowledge. The “instability” of some biodiversity indices due to new scientific knowledge means that conservation recommendations based on these indices should evolved along with the research on species taxonomy and phylogeny. We corrected the text to better highlight this point of view (lines 212-226): “This suggests that, contrary to HEDGE scores, LEDGE scores are only moderately robust to recent updates of the mammalian phylogeny. A possible explanation is that HEDGE scores seem to be more driven by extinction risk than LEDGE scores (Fig. S3), and therefore less impacted by phylogenetic changes. With the rapidly evolving phylogenetic knowledge, results will have to be regularly updated and recommendations may evolve accordingly – as this has been suggested and/or done in other studies dedicated to PD conservation³⁸⁻⁴⁰. In this regard the robustness of HEDGE to phylogeny updates is encouraging because it suggests that priority species will mainly remain the same, even with new phylogenetic knowledge. However, the moderate robustness of LEDGE to phylogenetic updates suggests that the identity of loss-significant species may change with new phylogenetic knowledge. Such instability reflects the evolution of scientific knowledge and likely does not jeopardise the survival of loss-significant species. This is because they are typically secure species, so new phylogenetic knowledge will help to better flag them before they become threatened.”

Nonetheless: (1) the relative instability of LEDGE scores for mammals does not make the approach of net positive outcome for PD less powerful from the conceptual point of view and (2) the first side of the approach of net positive outcome for PD, consisting in maximising gains in expected PD (ie the HEDGE approach), can be considered stable. We reworded the conclusion to make this clearer (lines 458-471): “The strategy we used to identify loss-significant species and areas that would limit losses in expected PD is only moderately robust

to recent phylogenetic updates, indicating that the identity of these loss-significant species and areas may need to be updated with new phylogenetic knowledge. Nonetheless, implementing those strategies in multiple taxonomic groups, i.e. maximising gains and minimising losses of expected PD, is a proactive way to reach a conservation target of net positive outcomes for PD. Because it reflects the positive effects conservation actions can have on expected PD, such a conservation target offers a more positive vision for the future than imagining the consequences of further PD losses only. This may therefore help to widen the engagement of decision-makers in PD conservation. We thus hope that our work will, together with other initiatives such as the IUCN Species Survival Commission Phylogenetic Diversity Task Force¹⁷, inform the CBD's Scientific Subsidiary Body and encourage the inclusion of a conservation target requiring net positive outcomes for PD in the CBD post-2020 framework.”

R3.4. To continue in this vein: in my first review, I highlighted that the nominal $p(\text{ext})$ (here, in a 50 year window) that is attached to a IUCN category is not in fact the $p(\text{ext})$ of a species assessed as belonging to that category, at least not if one considers actual change in species trajectories to extinction. The authors' reply, that this does not matter because one can just define these $p(\text{ext})$ to represent the situation if nothing were to be done from here on is technically correct, except that much is being done. This matters when policymakers are courted, actual species lists are presented, and particular species are highlighted. As an academic exercise, it would be very useful to see how robust HEDGE scores are to constant rank but changing $p(\text{ext})$ values; in a paper explicitly directed to the folks who might influence framers of international conventions, I believe it is critical. We simply do not know if the HEDGE lists and HEDGE areas are robust to the $p(\text{ext})$ used. (I do not believe we know this for EDGE lists either). I did not see any substantive response to this in the new version. I draw the authors attention to one reasonable way forward, most recently suggested by Gouhier and Pillai (Frontiers in Ecol and Evol, 2020), that phylogenetic information be used as an index to rank species within IUCN categories.

Reply to R3.4. We disagree with reviewer 3 requirement that we test the robustness of our framework to estimations of probabilities of extinction, specifically because this paper is policy-facing. We hope that our arguments below will convince him.

We disagree because, by definition, the indices HEDGE and LEDGE we use are dependent on the definition of $p(\text{ext})$ and are expected to change with this definition. The $p(\text{ext})$ we use indeed represents a baseline scenario in 50 years if nothing were to be done from here on to change species trajectories to extinction. Such baseline scenario does not reflect how ongoing conservation actions can change species trajectories to extinction. We agree with that point. Similarly the EDGE index, by construction, is dependent on the definition of the IUCN categories and of their estimation at a given time. It cannot consider how ongoing conservation actions can change species trajectories to extinction. Such a consideration would require models of species survival given environmental changes and ongoing conservation, models that currently do not exist at the mammalian phylogeny. The best that can be

suggested is to update the calculation of HEDGE, LEDGE, EDGE, ... metrics at each new update of the IUCN database.

What is policy-facing in this paper is actually the approach of net positive outcome applied to PD (independently of the taxonomic group we targeted, the phylogeny used and the $p(\text{ext})$ used). The HEDGE species & areas presented here are those to protect in priority to reach such net positive outcome for PD for the specific case of mammals. We acknowledged in the first revision of this manuscript that this approach of net positive outcome for PD can be applied using different methods to estimate $p(\text{ext})$. We have now improved this revision to better acknowledge of the dependence of our framework on the definition of the probabilities of extinction (lines 434–450): “Finally, we used IUCN designations projected to 50 years to derive probabilities of extinction from Red List extinction risk categories⁵² for each species and calculate expected PD in 2070. It means that our baseline scenario for 2070 corresponds to the remaining PD we would observe if, from now on, nothing could change species trajectories to extinction. The species and areas of interest we defined correspond to those whose protection would maximise gains (priority species/areas) or minimise losses (loss-significant species/areas) in expected PD compared to this baseline scenario (Fig. 1). Our results are dependent on the model we chose to calculate species' probabilities of extinction. In particular it does not consider how conservation actions could alter these probabilities in the next decades. Further studies would be needed to investigate which existing model of species extinction is most suitable for concrete conservation actions – as, by definition, the HEDGE, LEDGE and other expected PD statistics are dependent on the definition of species' extinction probabilities. Further studies are also needed to improve current estimations of these probabilities. As new models are being developed to estimate probabilities of extinction, such as a recent one considering the past dynamics of species through IUCN Red List extinction risk categories⁵³, we note that our approach could be applied to any baseline scenario using such new models.”

Therefore, although this comment made us improve the manuscript by insisting on the meaning of the baseline scenario and the fact that alternative models could be used to calculate $p(\text{ext})$, we believe that testing our approach by varying the scenario used to calculate $p(\text{ext})$ values would considerably lengthen an already dense paper without major benefits for its take-home message: (i) targeting a net PD outcome could help to widen the engagement of decision-makers in PD conservation, (ii) we identified the species and areas to conserve to reach this net PD outcome for the case of mammals and (iii) such newly-identified species and areas of interest currently lack protection.

R3.5. I can now turn to a few additional, new issues.

There are over 3000 species with reported HEDGE scores = 0.000, but these are given different ranks (ranging from 2200-5476.5); this oddity (or pathology) can be seen in the discontinuity in Figure S4 and the odd HEDGE-HEDGE plot in S1. These are likely zero-

length pendant edge species in the Phylacine tree, but of course no mammal species is zero years old. This needs to be dealt with. The fact it was not suggests this paper was a bit rushed.

Reply to R3.5. These species with reported HEDGE scores = 0.000 are actually species with HEDGE scores < 0.001. So this is not a problem due to zero-length pendant edge species in the Phylacine tree, but a problem of number of decimal places displayed in table S1. We corrected this and now display 8 decimal places for HEDGE scores in table S1. The minimum HEDGE score is 0.00000047 Ma for the two species *Sciurus igniventris* and *Sciurus spadiceus*, which are both given the rank 5476.5. Only species that have exact same HEDGE scores are given the same rank.

R3.6. In expectation, EDGE and HEDGE scores can be quite different, but whether they are empirically would seem important for situating this paper in the literature. If we drop these low-rank, incorrect zero-branch species, the simple correlation coefficient between EDGE and log(HEDGE) is ~0.80 (and same if one looks at the ranks only). But what if we look at only the top, say, 100 or 1000 species? And if high priority EDGE and high priority HEDGE species and areas of the world are not the same, which species or taxa or situations explain this?

Reply to R3.6. The Spearman rank correlation between HEDGE and EDGE scores is 0.94 as shown in Figure S4 (as specified above there is no need to remove the low-rank species because they are not incorrect zero-branch species). Further, as shown in table S2, 87 % of the TOP 25% HEDGE species (corresponding to the top 1369 species) are also in the TOP 25% EDGE species. We also give an example of a species that is in the TOP 25% HEDGE but not in the TOP 25% EDGE: the black-bearded flying fox *Pteropus melanopogon* (HEDGE rank = 820, EDGE rank = 1494). This species is among the least original but is endangered. Thus, despite the fact that it is not a priority EDGE species, managing to secure it would still bring more important gains in expected PD than securing any of the 4657 mammal species with a lowest HEDGE score.

We modified figure S5 to make a zoom on the relation between EDGE and HEDGE ranks for the TOP 25% HEDGE species, and discussed all of the above as follows (lines 288-300): “HEDGE and EDGE scores are highly correlated ($\rho = 0.94$) and most of the priority species (87%) are also species with the highest EDGE score (TOP 25% EDGE; Fig. S5, Table S3), i.e. the metric used to identify priority species by the EDGE of Existence Programme (www.edgeofexistence.org) – the only global conservation initiative to focus specifically on threatened species that represent a significant amount of unique phylogenetic diversity. Although HEDGE and EDGE metrics differ, this finding was expected as both give the highest scores to species that are very evolutionarily distinct and very threatened, and is consistent with previous findings⁴⁵. However, among the bottom half of the TOP 25% HEDGE species, some do not belong to the TOP 25% EDGE, such as the black-bearded flying fox *Pteropus melanopogon*. This species is among the least original but is endangered. Thus, despite the fact that it is not a priority EDGE species, managing to secure it would still

bring more important gains in expected PD than securing any of the 4657 mammal species with a lowest HEDGE score.”

R3.7. These questions take me back to the first political issue I raised above: given EDGE is the only active diversity-index conservation ranking system in place globally, and that it was introduced using mammals, it would seem natural that it be the baseline for considering HEDGE (and LEDGE) scores. As the authors know well, there are many diversity indices littering the literature. We need good reasons to start dropping those that are redundant, unstable, hard to communicate, or that do not capture aspects of biodiversity that matter. (I note to the editor that this may be completely out of line.)

Reply to R3.7. We acknowledge that because EDGE is currently the only metric for which there is an active conservation programme (Edge of Existence), EDGE is the baseline to which compare our HEDGE scores (but not the LEDGE scores which highlight relatively secure species). This is what we do in this manuscript, and we modified the text to make it clearer (lines 288-292): “HEDGE and EDGE scores are highly correlated ($\rho = 0.94$) and most of the priority species (87%) are also species with the highest EDGE score (TOP 25% EDGE; Fig. S5, Table S3), i.e. the metric used to identify priority species by the EDGE of Existence Programme (www.edgeofexistence.org) – the only global conservation initiative to focus specifically on threatened species that represent a significant amount of unique phylogenetic diversity.”

However, the indices we use here to identify priority and loss-significant species/areas are not redundant with EDGE because they are explicitly based on gains and losses in expected PD that our conservation actions would trigger. Although we show that LEDGE scores may be instable in case of changes in phylogenetic estimations, this is also the case for EDGE scores (see refs 13, 38, 39). In contrast, we found that HEDGE scores are quite stable. We recently learnt that such expected PD framework is being adopted by the Edge Of Existence programme (see <https://www.edgeofexistence.org/blog/cutting-edge-updating-science-behind-species/> + refs 16 & 18 in our manuscript, and this was further confirmed to us by one of the ZSL researcher working on the topic). Therefore, we expect that our approach and the approach adopted by this programme will converge in a near future.

There is now an abundant literature explaining that conserving PD is important, not only for its intrinsic values, but also because it represents a reservoir of yet-to-be discovered benefits for humanity. Therefore, the indices we use are related to biodiversity aspects which matter. Also the indices we use in our study are actually easy to communicate – specifically because they are directly related to conservation (in)actions. Priority species or areas are those whose protection would maximise gains in expected PD, and loss-significant species are those whose extinction would maximise losses in expected PD.

R3.8. I note that the link in Ref. 16 (Owen et al. proposal to CBD) does not work. I do not know what this endeavour represents.

Reply to R3.8. Thank you for noticing, we corrected the link to ref 16 in the manuscript, which is now correct:

<https://www.cbd.int/api/v2013/documents/6445B22E-1BA7-18B7-6D28-61A95052E841/attachments/IUCN-6.docx>

R3.9. 148: identified, not defined

Reply to R3.9. We corrected the text accordingly.

R3.10. 205: I have never seen the term "picturing" used like this – please check.

Reply to R3.10. We changed it to the more common but synonym term “imagining”.

R3.11. 220 (and elsewhere). This analysis of introduced species seems completely off-topic, especially if the goal is to engage with policy makers. It is a distraction and should be toned down or dropped.

Reply to R3.11. We dropped the parts of the text and of figure 2 related to the analysis of introduced species.

R3.12. Overall, some of the new text reads as if it was rushed; it could be tightened.

Reply to R3.12. Dropping the parts on introduced species contributed to tighten the text. We also made minor text edits to improve the readability.

R3.13. Finally, I apologize for the extreme tardiness of this review. I have no good excuse except pandemic-induced administrative hassles with my lab and my students, and lower productivity overall. I appreciate that the authors may disagree with some of my comments, but I hope they are helpful nonetheless.

Arne Mooers

Reply to R3.13. We thank Reviewer 3, Arne Mooers, for his comments. Although we still disagree with two of them after this second round of review (R3.4 and R3.7), we modified our paper according to all comments (including R.3.4 and R.3.7) and gave full consideration of these comments in our revision. Overall, all comments have been useful and helped us to improve the quality of this manuscript.

REVIEWERS' COMMENTS

Reviewer #3 (Remarks to the Author):

While I still have some reservations, for example concerning the LEDGE metrics, I am comfortable recommending acceptance of this revised version pending a few last issues that I outline in the attached .pdf version, where I highlight the relevant bit and add in a Note in Preview (MacOS). There are about 20.

Apologies for the tardiness of this second review.